# Oscillatory brain activity in spontaneous and induced sleep stages in flies

Melvyn H.W. Yap[1], Martyna J. Grabowska[1], Chelsie Rohrscheib[1], Rhiannon Jeans[1], Michael Troup[1], Angelique C. Paulk[1,2], Bart van Alphen[1,3], Paul J. Shaw[4] & Bruno van Swinderen[1]

Sleep is a dynamic process comprising multiple stages, each associated with distinct electrophysiological properties and potentially serving different functions. While these phenomena are well described in vertebrates, it is unclear if invertebrates have distinct sleep stages. We perform local field potential (LFP) recordings on flies spontaneously sleeping, and compare their brain activity to flies induced to sleep using either genetic activation of sleep-promoting circuitry or the GABA$_A$ agonist Gaboxadol. We find a transitional sleep stage associated with a 7–10 Hz oscillation in the central brain during spontaneous sleep. Oscillatory activity is also evident when we acutely activate sleep-promoting neurons in the dorsal fan-shaped body (dFB) of *Drosophila*. In contrast, sleep following Gaboxadol exposure is characterized by low-amplitude LFPs, during which dFB-induced effects are suppressed. Sleep in flies thus appears to involve at least two distinct stages: increased oscillatory activity, particularly during sleep induction, followed by desynchronized or decreased brain activity.

[1] Queensland Brain Institute, The University of Queensland, St Lucia, QLD 4072, Australia. [2] Department of Neurological Surgery, Massachusetts General Hospital, Harvard Medical School, Boston, MA 02114, USA. [3] Department of Neurobiology, Northwestern University, Evanston, IL 60208, USA. [4] Department of Anatomy & Neurobiology, Washington University School of Medicine, St Louis, MO 63110, USA. Correspondence and requests for materials should be addressed to B.v.S. (email: b.vanswinderen@uq.edu.au)

The sleeping brain is far from quiet in most animals where it has been studied carefully, displaying distinct forms of brain activity accomplishing potentially different functions[1]. These sleep stages are typically associated with electrophysiological signatures. Slow wave sleep (SWS), for example, is characterized by 1–4 Hz activity, and these widespread brain oscillations have been proposed as a mechanism for downscaling synapses[2] or for clearing metabolites from the

brain[3]. SWS epochs alternate with rapid-eye movement (REM) sleep, which is characterized by wake-like brain activity and has been linked to other functions, such as memory consolidation and motor learning[4, 5]. These dynamic sleep processes were originally believed to be unique to mammals and birds, but recent work in reptiles suggests that SWS-REM alternations may have evolved much earlier[6]. It is unclear if other animals such as invertebrates display similar dynamic processes during sleep, in part because criteria such as REM are not useful for animals lacking the capacity to move (or close) their eyes. However, it is evident that even insects sleep[7, 8] and work in *Drosophila* flies suggests that some proposed sleep functions, such as synaptic downscaling and memory consolidation, are conserved across species[9, 10]. More recent work in *Drosophila* has shown that behavioral responsiveness can vary throughout a sleep bout[11], suggesting that even the smallest animal brains might display distinct sleep stages. Thus, SWS and REM sleep in reptiles, birds, and mammals might reflect a more ancient need for all brains to alternate between different sleep stages to potentially achieve distinct sleep functions[12].

Sleep has traditionally been studied in animals as a spontaneous behavior driven by interacting circadian and homeostatic processes[13, 14]. Recent genetic advances using the *Drosophila* model now permit sleep duration to be exquisitely controlled, by transiently activating sleep-promoting neurons[10, 15], thereby allowing hypothesized sleep functions to be tested experimentally. For example, sleep induction in flies has been found to improve learning in mutant animals[16] and this seems to be associated with altered synaptic physiology[17]. It remains unclear, however, if experimentally induced sleep in flies resembles any particular natural sleep stage. Whereas different approaches (genetic or pharmacological) have been used to induce sleep in *Drosophila*, fly sleep has typically been viewed as primarily a single process associated with extended quiescence[7, 8].

In this study, we recorded local field potentials (LFPs) from spontaneously sleeping flies and we characterize what appear to be different sleep stages, based on the LFP. To better understand these potentially distinct sleep stages, we compare sleep-induction effects achieved by two different ways: by transiently activating sleep-promoting neurons of the dorsal fan-shaped body (dFB)[10] and by exposing flies to Gaboxadol[18], a drug that increases SWS in humans. We then compare behavioral effects of induced sleep using either Gaboxadol or dFB activation, or both methods combined. We find that that either method recapitulates some aspects of spontaneous sleep, such as increased oscillatory activity at the beginning of a sleep bout, or decreased overall LFP activity in the middle of sleep bout. While both experimental approaches produce a similar level of sleep intensity, the behavioral consequences are different for extended sleep using either method alone. Our study suggests that sleep initiation in flies is an active brain process distinct from other forms of fly sleep, which argues that different sleep stages already emerged in the smallest animal brains.

## Results

**Oscillatory brain activity during spontaneous sleep in flies.** We first investigated the neural correlates of spontaneous sleep in wild-type flies. Previously, we have shown that sleep in *Drosophila* is associated with, on average, decreased LFP activity compared to wake[11, 19, 20]. As before, we recorded LFPs by implanting two glass electrodes into both brain hemispheres (see Methods section) and extracting an amplified voltage differential[11]. We improved the behavioral context of our overnight recording setup by placing tethered flies on an air-supported ball, and continuously filming our experiments day and night under infrared lighting (Fig. 1a). Flies slept readily in this context, with several flies displaying extended sleep bouts of up to 20 min (Supplementary Fig. 1a). To confirm that flies were indeed asleep and not just awake and immobile, we periodically applied a mechanical stimulus to test for arousal (Fig. 1a, b), and determined whether flies responded (by walking on the ball) within 15 s (see Methods section). Flies that were immobile for over 5 min were significantly less responsive than flies that were immobile for less than 1 min (Fig. 1b, bottom panel). This established our sleep criteria for this tethered recording preparation (>5 min immobility), which agrees well with behavioral work in the field[7, 8].

We used wavelet analysis[21] to examine how LFP frequencies changed through time, across 24 h of wake and sleep. As found previously[11, 19, 20], sleep in flies is associated with overall decreased LFP activity (Fig. 1c, d; Supplementary Fig. 1b). However, wavelet analysis also revealed a marked ~8 Hz oscillation (and associated harmonics) in several flies, especially during sleep (Fig. 1c, white arrows, and zoomed in panels on the right; Supplementary Fig. 1b). This oscillation was largely absent during wake (Fig. 1c and Supplementary Fig. 1b). Crucially, the oscillation was not present in awake yet immobile flies (Fig. 1c, right zoomed in panels), and seemed of variable intensity throughout a sleep bout (Fig. 1c, middle zoomed in panels; Supplementary Fig. 2)—ruling out the likelihood of an artifact linked to postural quiescence on the air-supported ball. Also, the oscillation was not an artifact of micro-behaviors, such as grooming and proboscis extension (Supplementary Fig. 2), and was not an artifact of the fly's heartbeat (see Methods section). Rather, the oscillation appeared intermittently mostly in immobile, sleeping flies—day or night (Fig. 1e, bottom), and was not as prominent in awake flies—day or night (Fig. 1e, top). Since the oscillation's frequency could vary among and even within animals (Supplementary Fig. 2), in subsequent analyses, we defined it as 7–10 Hz.

**Fig. 1** Increased 7–10 Hz oscillations during spontaneous sleep. **a** In vivo overnight LFP recording setup (see Methods section). **b** Behavioral responses to a mechanical stimulus, in relation to prior immobility time. Top: three sample traces. Colored bars on the x-axis indicate the time period bins used for calculating response proportion. Bottom: Average response ($\pm$ s.e.m.) for four prior immobility durations ($n = 7$, *$p < 0.05$, **$p < 0.01$ by Friedman test with Dunn's multiple comparisons between all immobility durations). **c** Left: spectrogram of LFP amplitude (0–40 Hz power, see Methods section) of a sample fly recording over 24 h (top), with corresponding raw LFP signal (middle) and behavioral activity quantified as pixel changes (bottom). Right panels show expanded views of a 5-min segment of a sleep epoch (black box) and a 5-min segment of a wake epoch (orange box). White arrows indicate some instances of ~8 Hz oscillations. **d** Average 0–100 Hz LFP power ($\pm$ s.e.m.) during nighttime and daytime sleep is significantly reduced compared to daytime wake ($n = 10$, *$p < 0.05$, ***$p < 0.001$ by Friedman test with Dunn's multiple comparisons between all conditions). **e** Average 6–10 Hz power spectra for sleep and wake states during day and night ($n = 10$ flies, same color code as in **d**). **f** Sleep bouts (> 5 min) were binned into 5 segments (1 min each, except for mid-sleep, which was variable in length) to compare LFPs from early to late sleep. **g** Average 7–10 Hz power ($\pm$ s.e.m.) for each sleep epoch, normalized to mid-sleep. ($n = 10$ flies, *$p < 0.05$, **$p < 0.01$ by Friedman test with Dunn's multiple comparisons between each sleep segment and mid-sleep). Images: Melvyn Yap

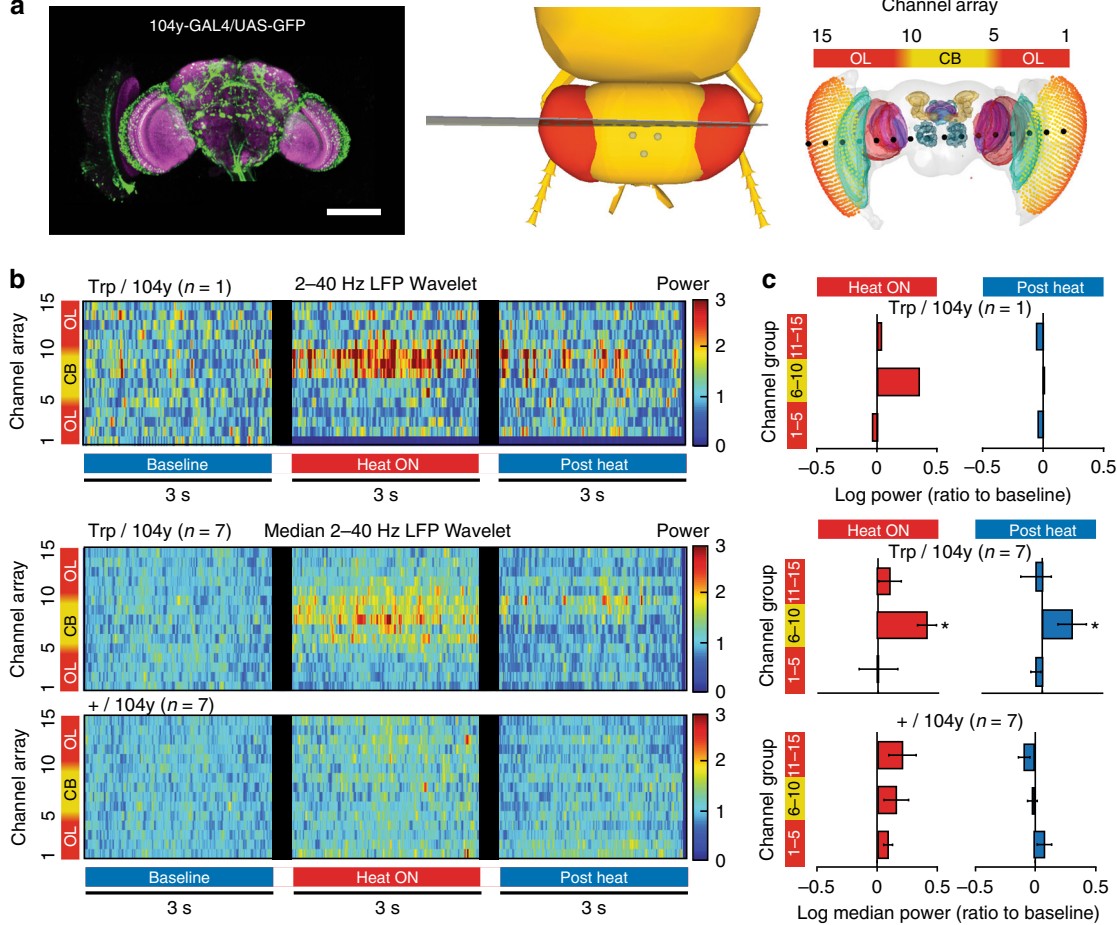

**Fig. 2** Thermogenetic sleep induction increases 2–40 Hz power in the central brain. **a** Left image shows the expression pattern of 104y-Gal4 circuit as visualized by green fluorescent protein (GFP) expression (green). The synaptic marker nc82 highlights neuropil structures (magenta). Middle and right panels indicate the approximate locations of the 16 channel probe (15 functional channels) in a standardized *Drosophila* brain[21]. The channels were grouped into three regions: two optic lobes region (left and right, OL), and the central brain (CB). Scale bar = 100 μm. **b** Spectrograms show 2–40 Hz LFP power across all 15 channels (grouped as OL and CB) in 3-sec segments associated with heat conditions and then concatenated, for an individual fly expressing 104y-Gal4/UAS-TrpA1 (Trp/104y, top), the combined median power for Trp/104y (n = 7, middle), and 104y-Gal4/+ (+/104y, n = 7, bottom). Blue and red bars at the bottom indicate the temperature stimulus. **c** The 15 recording channels were grouped into the 3 aforementioned regions for the purpose of analysis. Bar plots shows the median 2–40 Hz LFP power during the periods of Heat ON (red) and Post Heat (blue) relative to values at baseline, which was normalized to zero. Comparisons were made for an individual UAS-TrpA1/104y-Gal4 fly (top), the combined averaged power for UAS-TrpA1/104y-Gal4 (middle, n = 7, *p < 0.0125), and 104y-Gal4/+ controls (bottom, n = 7, ns). UAS-TrpA1/+ controls were also tested (not shown) and no significant effects of heating were found (n = 7). Statistical significance was determined by multi-factor ANOVA with post hoc contrasts on a three-way interaction term between brain regions, fly line, and heat condition (Supplementary Note 1 and Supplementary Tables 1 and 2). Sample sizes indicate the number of flies tested. Images: Angelique Paulk

Since 7–10 Hz activity often appeared intermittent, we wondered whether the oscillation was more prominent at the beginning or end of the night, as this might suggest homeostatic regulation. To address this, we divided all spontaneous sleep bouts into three equal epochs per fly: early-night sleep, mid-night sleep, and late-night sleep (see Methods section). Comparisons of normalized LFP power between early and late sleep showed no significant differences (Supplemental Fig. 1c), although there was more variability in 7–10 Hz power early in the night. The observation that 7–10 Hz power is equally prominent during sleep at the beginning of the night as it is later in the night was also visually evident in individual spectrograms, e.g., Fig. 1c.

We next questioned if the 7–10 Hz oscillations changed in amplitude within a single sleep bout. To address this, we partitioned all sleep bouts (>5 min) into five segments, to capture early sleep LFP activity (0–2 min after quiescence onset), mid-sleep activity (of variable duration, >1 min), and sleep prior to spontaneous awakening (0–2 min prior to first movement, Fig. 1f).

We found that 7–10 Hz oscillations were significantly more pronounced during early sleep and immediately prior to awakening, compared to the middle of a sleep bout, for both daytime and nighttime sleep (Fig. 1g). This effect was not significant for higher-frequency domains (50–100 Hz), although other lower-frequency domains (e.g., 2–6 Hz, 15–30 Hz) also showed this pattern to some extent (Supplementary Fig. 3). These results suggest that LFP oscillations, especially in the lower frequencies (<50 Hz) are associated with a distinct transitional sleep stage soon after sleep onset, or within 1 min prior to awakening. This pattern suggests a function linked with promoting transitions between sleep and wake.

We were curious whether flies engaged in 7–10 Hz sleep were more responsive to stimuli because we had previously found that flies could be more easily aroused during early stages of sleep[11]. We recorded from flies that were regularly stimulated throughout the night, and then identified epochs of high vs. low 7–10 Hz activity that coincided with a mechanical stimulus (see Methods

section). We found no significant difference between behavioral responsiveness in either group (Supplementary Fig. 1d). This suggests that 7–10 Hz sleep in flies is not equivalent to 'lighter' sleep, at least in the context of our tethered recording preparation.

**Oscillatory brain activity during induced sleep in flies.** Sleep can be artificially induced in *Drosophila* by activating neurons that innervate the dFB in the central brain of the fly[10, 15, 22]. In addition to producing behavioral quiescence, thermogenic dFB

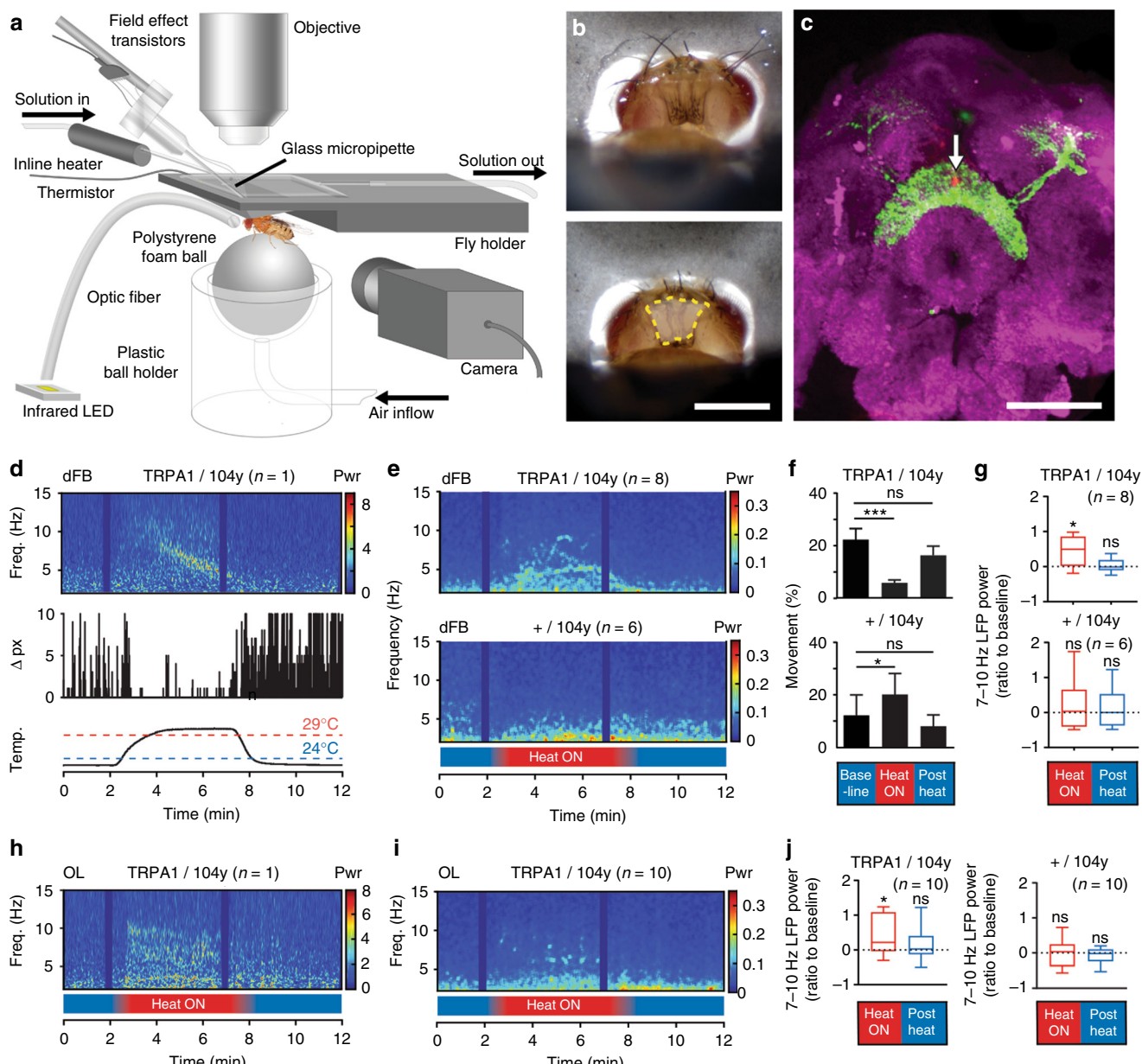

**Fig. 3** Oscillations of 7–10 Hz induced in dFB. **a** In vivo exposed-brain LFP recording setup optimized for thermogenetics (see Methods section). **b** Close-up showing the posterior head (top), and with part of cuticle removed exposing the brain (bottom). Scale = 0.5 mm. **c** GFP expression of 23E10-Gal4 (green) highlighting the dFB. nc82 (magenta) highlights neuropil. Arrow, LFP recording site. Scale = 50 μm. **d** Spectrogram of 2–15 Hz LFP power (top) for a 12-min LFP recording in the dFB in 104y-Gal4,UAS-mCD8::GFP/UAS-TrpA1 (TRPA1/104y), with corresponding behavioral activity (Δ pixel, middle) and brain perfusion temperature (bottom). Dark vertical bars are excluded artifact. **e** Spectrograms of averaged 2–15 Hz LFP power for dFB recording in TRPA1/104 y flies (n = 8, top) and control strain 104y-Gal4,UAS-mCD8::GFP/+(+/104 y, n = 6, bottom). Bottom bar indicates time when bath temperature exceeded 29 °C (Heat ON, red). **f** Average percentage (± s.e.m.) of time fly spent moving during baseline, Heat ON, and Post Heat conditions for TRPA1/104y flies (top, n = 22, by Friedman test with Dunn's multiple comparisons), and for +/104y flies (bottom, n = 17, *p < 0.05, Friedman test with Dunn's multiple comparisons). **g** Median 7–10 Hz LFP power in the dFB during Heat ON for TRPA1/104y flies (top, n = 8, *p < 0.05, one sample t test comparing to baseline of zero for Heat ON), and for +/104y flies (bottom, n = 6, ns, by one-sample t test comparing to baseline of zero for both Heat ON and Post Heat). **h** Spectrogram of 2–15 Hz LFP power for recording in the optic lobe (OL) of a sample TRPA1/104y fly. **i** Spectrogram of average 2–15 Hz LFP power for OL recording in TRPA1/104y flies (n = 10). **j** Left, median 7–10 Hz LFP power in the OL during Heat ON for TRPA1/104y flies (left, n = 10, *p < 0.05 by one sample t test comparing to baseline of zero for Heat ON condition, and ns by Wilcoxon signed rank test comparing to baseline of zero for Post Heat condition); right, median 7–10 Hz LFP power for +/104y flies (right, n = 10, ns by one sample t test comparing to baseline of zero for Heat ON and Post Heat). ns, not significant. Images: Melvyn Yap

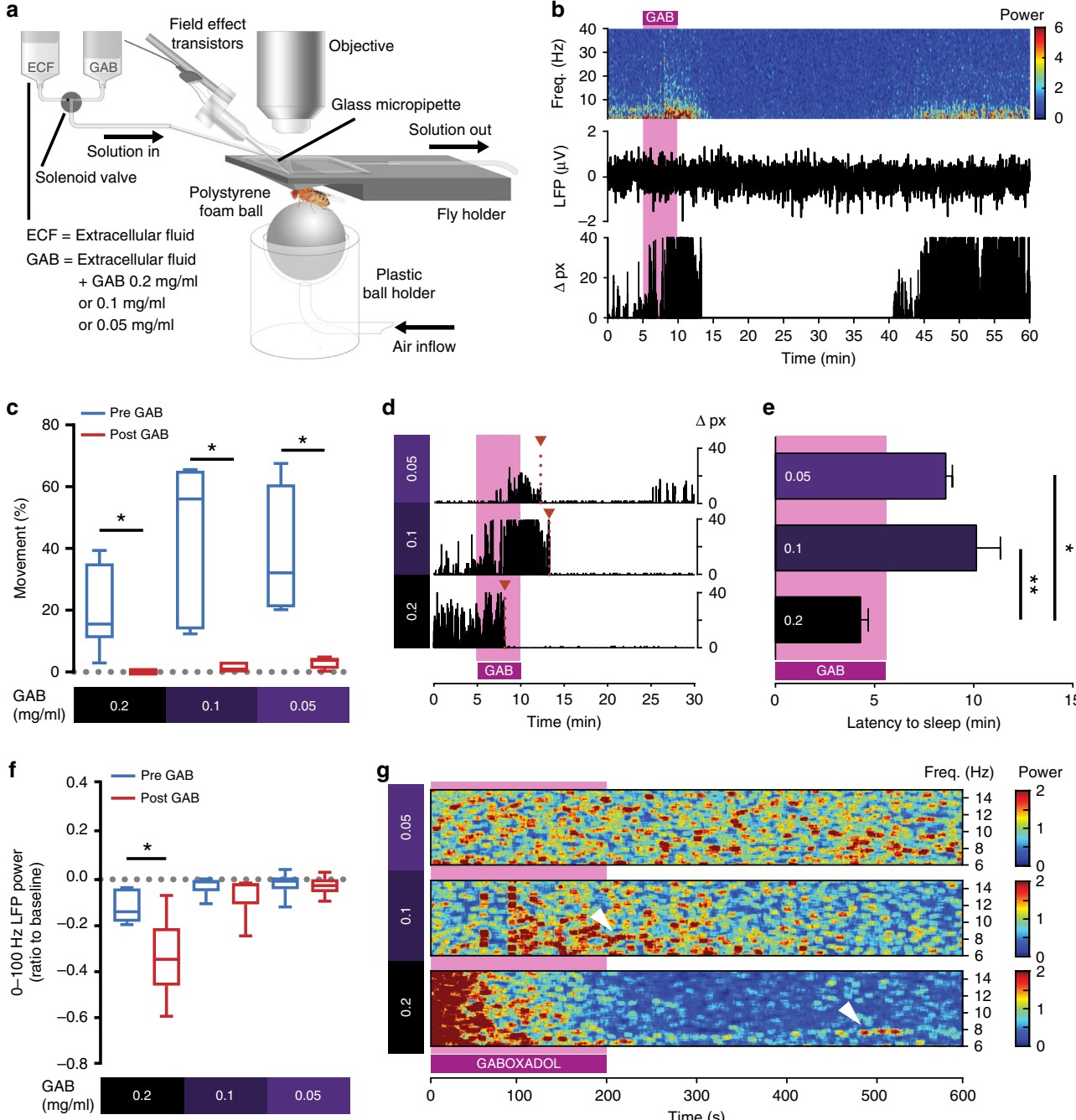

**Fig. 4** Gaboxadol-induced sleep is associated with an overall decrease in LFP activity. **a** In vivo exposed-brain LFP recording setup optimized for pharmacological experiments (see Methods section). **b** Spectrogram of 2–40 Hz LFP power (top) taken from the dFB of a fly exposed to Gaboxadol (0.1 mg/ml) for 5 min (magenta shade), with corresponding filtered LFP signal (middle) and corresponding behavioral activity quantified as pixel changes (bottom). **c** Median percentage of time fly spent moving within 5 min prior (blue) and 5 min after (red) the pre-determined drug onset (see Methods section) (*$p < 0.05$, by Wilcoxon matched pairs signed rank test between pre- and post-drug). **d** Representative fly movement (pixel change) for each concentration of Gaboxadol displaying latency to sleep for each Gaboxadol concentration (indicated on the left). **e** Averaged latency to sleep (± s.e.m.) was significantly earlier for flies exposed to a Gaboxadol concentration of 0.2 mg/ml ($n = 6$) compared to both 0.1 mg/ml ($n = 6$, **$p < 0.01$) and 0.05 mg/ml ($n = 6$, *$p < 0.05$ by Kruskal–Wallis with Dunn's multiple comparisons between all concentrations). **f** A significant decrease in the overall LFP power (0–100 Hz) was observed when flies were exposed to Gaboxadol 0.2 mg/ml ($n = 6$) but not for lower concentrations (*$p < 0.05$ by Wilcoxon matched pairs signed rank test between pre- and post-drug). **g** Spectrograms of individual fly LFP recordings starting 1 min after the onset of Gaboxadol perfusion, for three concentrations of Gaboxadol. White arrows indicate 7–10 Hz oscillations. Sample sizes indicate the number of flies tested. Images: Melvyn Yap

activation has been shown to promote a key sleep function, memory consolidation[10]. Since spontaneous sleep initiation appears to be associated with increased LFP activity, especially in the lower frequencies, we wondered whether dFB activation would similarly display a sleep-related LFP signature. To control sleep on demand, we expressed a temperature-sensitive cation channel, TrpA1[23], in a dFB-expressing circuit that has been previously shown to promote sleep, 104y-Gal4 (Fig. 2a, left panel)[10, 22]. Sleep was achieved by increasing the temperature of 104y-Gal4/ UAS-TrpA1 flies to >29 °C (see Methods section). To record brain activity from sleeping flies, we used a multichannel preparation that samples LFPs from 16 channels simultaneously across the *Drosophila* brain (Fig. 2a, middle and right panels)[21]. Transient circuit activation using the same thermogenetic approach in this multichannel preparation has previously uncovered distinct oscillations across the waking fly brain[21], but the electrophysiological effect of activating sleep-promoting circuits has never been investigated. Recording from multiple sites simultaneously should thus reveal any changes in the LFP during sleep induction, and identify roughly where in the brain these occur.

Consistent with our spontaneous sleep recordings, we found that sleep induction achieved by thermogenetic activation of 104y-Gal4 circuits is associated with increased LFP activity in our multichannel recordings, although across a broader frequency range (2–40 Hz) (Fig. 2b, c, an individual example is shown in the top row, median data in the middle row; Supplementary Fig. 4a). Increased LFP activity upon sleep induction was nevertheless surprising because sleep is generally associated with decreased LFP amplitudes in insects and other invertebrates[19, 20, 24–26]. We noted that most increased LFP activity in 104y-Gal4/UAS-TrpA1 flies was in the central brain (Fig. 2c, middle row red bars), and this significant effect persisted after the heat was turned off (Fig. 2c, middle row blue bars). Closer examination across frequencies in the central brain revealed a wide range of effects, with prominent activity in the lower frequencies (Supplementary Fig. 4a). We confirmed this increased LFP effect using another sleep-promoting dFB-expressing line, C5-Gal4[27], which also showed increased LFP activity in the central brain during sleep induction (Supplementary Fig. 4b), but also showed spontaneous activity during baseline in some flies (Supplementary Fig. 4e and Supplementary Note 1, Multichannel recordings 2–40 Hz analysis). Interestingly, induced LFP activity was rarely confined to just one recording site or even just the central brain; rather, the increased LFP activity was often intermittent, and appeared to travel from one brain location to another, sometimes even impacting the optic lobes (see individual examples for 7–10 Hz activity specifically, in Supplementary Fig. 4c, e). In contrast, genetic controls showed no increased LFP activity on average during heating (Fig. 2b, c, bottom row, and see legend for UAS-TrpA1/ + data).

**Oscillatory brain activity is produced by dFB neurons**. To confirm that the source of these oscillations is indeed in the central brain, we employed a different, more focal recording preparation (Fig. 3a). We exposed the fly brain by opening the cuticle at the back of the head (Fig. 3b), and inserted a glass electrode directly into the dFB (guided by GFP expression and local dye release, Fig. 3c, see Methods section), from where we recorded LFPs. To induce sleep (in 104y-Gal4/UAS-TrpA1;UAS-GFP flies), we raised the temperature of the brain perfusion solution to >29 °C (Fig. 3d, bottom panel). Flies with their brain thus acutely heated promptly fell asleep as predicted (Fig. 3d, middle panel; Supplementary Movie), while controls stayed awake (Fig. 3f; Supplementary Movie). We again observed

prominent LFP oscillations associated with sleep induction (Fig. 3d, top panel; Supplementary Movie), as in our multichannel experiments on the same strains. Interestingly, the frequency and intensity of the oscillations could change through time, as we also saw during spontaneous sleep (Fig. 1c; Supplementary Fig. 2), although induced LFP activity recorded directly from the dFB was most prominent in the lower-frequency ranges (*e.g.*, 6–15 Hz) on average (Supplementary Fig. 5a), and also significant in the 7–10 Hz range (Fig. 3g, top). In contrast, heating the brains of control flies had no significant effect on LFP activity (Fig. 3e, g, bottom panels; Supplementary Fig. 5b; Supplementary Movie), although we noted considerable variance in the LFP, perhaps as a consequence of the control flies reacting to heat while awake. To check whether these induced oscillations spread beyond the central brain, we performed additional focal recordings from the optic lobes of 104y-Gal4/UAS-TrpA1 flies, and these also revealed LFP oscillations in some flies (e.g., Fig. 3h). These oscillations in the optic lobes were significant for the 7–10 Hz range, compared to baseline (Fig. 3j, left panel), which was not the case for control flies (Fig. 3j, right panel). Other frequency bands showed no significant change in the optic lobes (Supplementary Fig. 5c, d). As also revealed by our multichannel recordings (Fig. 2 and Supplementary Fig. 4), this suggests that dFB-associated sleep induction is associated with increased LFP activity that is most pronounced in the central brain, but that may also impact some other parts of the fly brain such as the optic lobes. While we did find some consistency in the 7–10 Hz range between spontaneous and genetically induced sleep, effects across a broader frequency range (2–40 Hz) were also evident using both approaches, suggesting some variability in the frequency domain for this sleep-related oscillation.

**No significant activity during drug-induced sleep**. An alternative approach to inducing sleep in *Drosophila* is to expose flies to a sleep-promoting drug, for example, the GABA agonist Gaboxadol, which was developed to treat insomnia[16, 28]. In humans, Gaboxadol has been shown to promote slow-wave (1–4 Hz) sleep and to suppress REM sleep[18]. Previous work has shown that Gaboxadol promotes spontaneous sleep in flies, and that, similar to dFB activation, Gaboxadol-induced sleep can also be restorative[16]. Rather than feeding Gaboxadol to flies, we adapted the exposed-brain preparation (Fig. 4a) to perfuse different concentrations of Gaboxadol directly to the brain while we recorded LFPs from the GFP-labeled dFB with sharp electrodes (Fig. 4b). We tested three concentrations of Gaboxadol (in mg/ ml): 0.05, 0.1, and 0.2. All three concentrations induced quiescence in flies (Fig. 4c), although the latency to quiescence was significantly shorter at 0.2 mg/ml (Fig. 4d, e), with this drug concentration achieving immobility within 5 min (Fig. 4e). Exposure to Gaboxadol was associated with decreased LFP amplitudes (Fig. 4f), resembling a later stage of spontaneous sleep. However, this effect was only reliably induced at the highest concentration tested (0.2 mg/ml, Fig. 4f; Supplementary Fig. 6). Whether the lower concentrations actually achieved sleep in all flies is questionable: when we exposed flies to 0.05, or 0.1 mg/ml of Gaboxadol, timing to quiescence was more variable (>10 min) and not usually associated with the expected overall decrease in LFP activity associated with invertebrate sleep (Fig. 4f, g and Supplementary Fig. 6). We therefore concluded that 0.2 mg/ml of the drug is the appropriate dosage for reliably inducing sleep in this preparation. Interestingly, LFP activity was not increased on average when flies were put to sleep by Gaboxadol, rather it decreased significantly across all frequencies, even for the 7–10 Hz range (Supplementary Fig. 6). However, we did observe some ~8 Hz LFP activity in two (out of 17) of the Gaboxadol

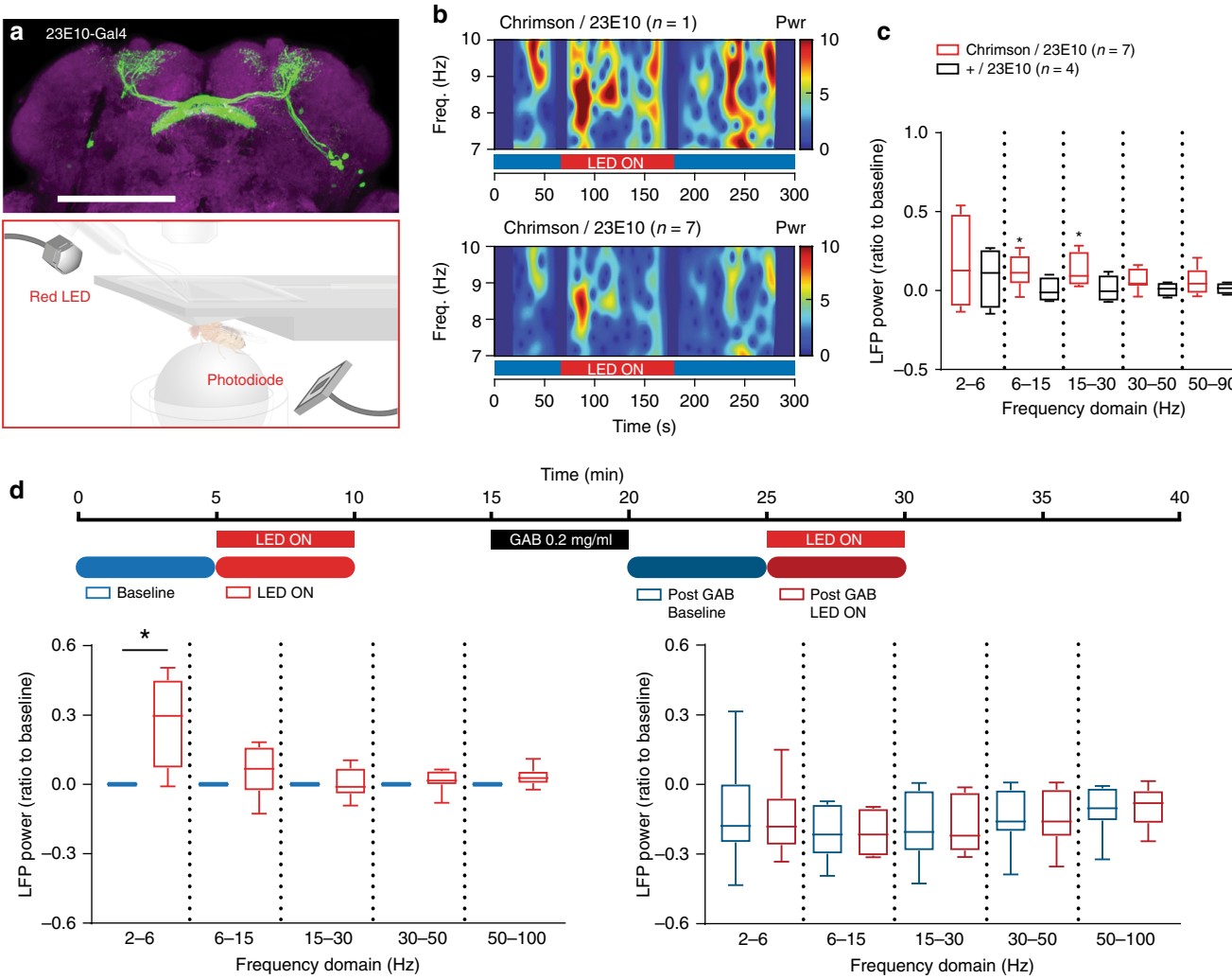

**Fig. 5** Low-frequency oscillations generated from activating the dorsal fan-shaped body (dFB) are abolished by Gaboxadol. **a** Target neurons for optogenetic activation (top), using the dFB-specific driver 23E10-Gal4 as visualized by green fluorescent protein (GFP) expression (green). Scale bar = 100 μm. In vivo exposed-brain LFP recording setup as in Fig. 3a, optimized for optogenetic experiments (bottom, see Methods section). **b** Spectrogram of a fly expressing 23E10-Gal4/UAS-CsChrimson (Chrimson/23E10) showing the presence of 7–10 Hz oscillation when dFB was optogenetically activated (top) and the averaged spectrogram (n = 7, bottom). Blue vertical bars represent excluded data due to the presence of external artifact. Bottom bar indicates when the light stimulus was on for all experiments (LED ON, red). **c** Photostimulation of CsChrimson-expressing dFB neurons was associated with a significant increase in the averaged 6–15 Hz and 15–30 Hz LFP power in the dFB (orange, n = 7, *p < 0.05 by Wilcoxon signed rank test comparing to baseline of zero), while this effect was not observed in the control strain 23E10-Gal4/+ (+/23E10, black, n = 4, ns by Wilcoxon signed rank test comparing to baseline of zero). **d** Top: experimental timeline indicating the time point of delivery of optogenetic stimulation, once prior to the delivery of Gaboxadol (0.2 mg/ml), and repeated once thereafter. Prior to exposure to drug, there was a significant increase in the average 2–6 Hz LFP power when optogenetically stimulated (bottom left, n = 8, *p < 0.05, by Wilcoxon matched pairs signed rank test between pre and post drug). No significant changes to the average LFP power of any frequency domain were detected with optogenetic stimulation after the drug was delivered (bottom right, n = 8, ns, by Wilcoxon matched pairs signed rank test between pre and post drug). Sample sizes indicate the number of flies tested. Images: Melvyn Yap

experiments (Fig. 4g, white arrows), showing that the alternate (dFB-mediated) sleep stage remained possible following 0.2 or 0.1 mg/ml drug exposure. Nevertheless, our results suggest that dFB-induced sleep is qualitatively different from Gaboxadol-induced sleep, as LFP activity was never significantly increased during sleep induction with the insomnia drug. Instead, Gaboxadol appears to promote a direct entry into a later stage of sleep characterized by overall decreased brain activity, without the dynamics observed previously (Supplementary Fig. 7), and potentially bypassing or suppressing the dFB-associated stage we have described previously.

**A role for the sleep switch in increasing LFP activity**. One way to explain the differences that we have uncovered between the two sleep-induction approaches is that the dFB promotes a different form of sleep that occurs in most spontaneous sleep episodes, but occurs less reliably in the medicated condition. To confirm that the dFB is responsible for generating increased LFP activity upon sleep induction, we used a more restricted sleep-promoting driver, 23E10-Gal4[22], which expresses in only ~20 cells that project to the dFB[29] (Fig. 5a, top panel). We employed an optogenetic approach to activate these dFB neurons (Fig. 5a, bottom panel), using UAS-CsChrimson, which is responsive to red light[30]. As for our 104y-Gal4/UAS-TrpA1 results, optogenetic activation of 23E10 neurons also resulted in increased LFP

activity in the dFB, especially in the lower-mid-frequency ranges (6–15 Hz and 15–30 Hz, Fig. 5b, c), and the flies slept (Fig. 6c). However, unlike our 104y-Gal4/UAS-TrpA1 results and our spontaneous sleep data, we never observed a distinct narrow-band oscillation. This nevertheless confirms that acutely activating dFB neurons does indeed increase LFP activity in the fly brain, regardless of the methods or reagents used.

Since dFB-induced sleep and Gaboxadol-induced sleep have opposing effects on LFP activity, we decided to combine both sleep-induction manipulations to determine whether the dFB-induced oscillations are suppressed by Gaboxadol, or whether they can still be produced after Gaboxadol-induced sleep. Sustained dFB-induced oscillations during Gaboxadol-induced sleep might suggest that these oscillations are an epiphenomenon unrelated to sleep, and perhaps simply associated with activating

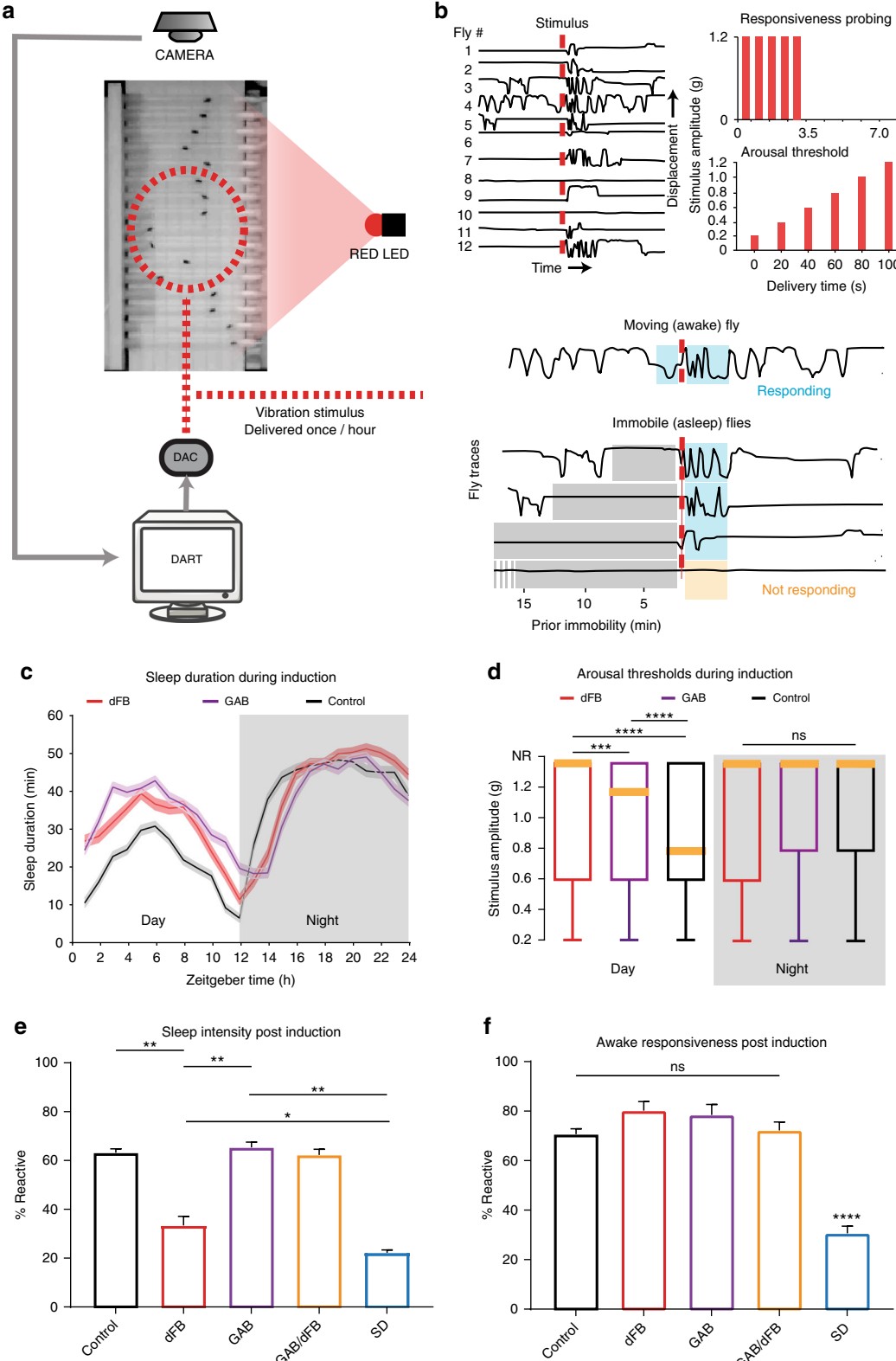

the dFB artificially. To test this, we exposed 23E10-Gal4/UAS-CsChrimson flies to 0.2 mg/ml Gaboxadol, observed them falling asleep, and then activated the dFB with red light (Fig. 5d). We first confirmed that dFB activation increased LFP activity (Fig. 5d, left panel), although significance was only evident in the lowest frequency range in this dataset. We then confirmed that perfusion of 0.2 mg/ml Gaboxadol decreased LFP power across all frequencies (Fig. 5d, right panel, blue). Flies remained asleep when the dFB was optogenetically activated during drug perfusion, and LFP activity remained suppressed (Fig. 5d, right panel, red). Although it is unclear why only a lower-frequency range (2–6 Hz) was significant in this data set, it is clear that Gaboxadol suppressed the dFB effect that was evident in the same flies. This shows that inducing sleep by potentiating GABA$_A$ circuits in the fly brain overrides the LFP oscillations that are associated with dFB-induced sleep.

**Behavioral effects of distinct sleep manipulations**. Since brain activity is clearly different following Gaboxadol and dFB activation, we questioned whether forced sleep using these two different methods might have distinct behavioral consequences. Our behavioral tracking methods confirmed that flies indeed became quiescent following either manipulation (Figs 3f and 4c), but these acute manipulations did not reveal whether sleep duration or intensity might be different for either sleep-induction method. We used the *Drosophila* ARousal Tracking (DART) platform[31] to probe sleep and behavioral responsiveness in freely walking flies exposed to either treatment (Fig. 6a). In addition to providing sleep duration metrics, DART measures responsiveness to mechanical stimuli in both sleeping and awake animals (Fig. 6b). Volleys of stimuli delivered every hour reveal sleep intensity data (as percent of flies responding) or arousal threshold data (as the stimulus intensity required to wake a fly up). Importantly, responsiveness can also be measured during wake (see Methods section), providing a functional readout for our sleep manipulations. To best compare our distinct sleep manipulations (dFB sleep vs Gaboxadol sleep), we used the same strain (23E10-Gal4/UAS-CsChrimson, as in Fig. 5) that had been fed either 0.1 mg/ml Gaboxadol or 0.5 mg/ml all-trans-retinal (ATR), and exposed these flies and vehicle-fed controls to red light. We found that both sleep-induction methods increased sleep duration to a similar extent, increasing daytime sleep (a ceiling effect was evident for nighttime sleep) (Fig. 6c and Supplementary Fig. 8a, b). There was no difference between the dFB and GAB experimental groups, however, both groups had significantly increased sleep over 24 h compared to the controls ($F(2,69) = 4.48$; dFB vs. GAB, $p = 0.16$; dFB vs. Control, $p = 0.02$; GAB vs. Control, $p = 0.03$, ANOVA with Tukey's multiple comparisons test). Both methods

also increased arousal thresholds (AT) during the day ($H(2,5289) = 164.0$; dFB vs. GAB, $p = 0.0002$; dFB vs. Control, $p < 0.0001$; GAB vs. Control, $p < 0.0001$), while all flies (including controls) were equally unresponsive during the night (Fig. 6d) (NR) $H(2,4388) = 1.96$; dFB vs. GAB, $p = 0.99$; dFB vs. Control, $p = 0.42$; GAB vs. Control, $p = 0.44$. This shows that both sleep manipulations promote a depth of sleep that is similar in intensity to spontaneous sleep at night, although we noted that dFB-activated flies were even less responsive during than Gaboxadol-fed flies during the day. The depth of sleep induced by dFB activation is consistent with our earlier observation that spontaneous 7–10 Hz sleep was not 'lighter' than other forms of sleep in flies (Supplementary Fig. 1d).

Next, we examined whether behavioral responsiveness was different after prolonged nighttime sleep via either manipulation. Our idea here was to replace spontaneous nighttime sleep with 12 h of Gaboxadol or dFB sleep. We then measured behavioral responsiveness throughout the subsequent 12-h day in two different ways: either as a proportion of flies responding (while asleep or awake) or as startle-induced locomotion speed (see Methods section). We found that dFB-activated flies slept more deeply than controls after 12 h of 'forced' dFB sleep at night, with only half as many responding compared to controls (Fig. 6e). In contrast, sleep intensity in Gaboxadol-fed flies was similar to controls, after they had been removed from the drug (Fig. 6e). Prolonged dFB sleep also appeared to make flies more sluggish upon waking, compared to Gaboxadol sleep (Supplementary Fig. 8c). We next asked what the effect would be if we combined both sleep induction methods, as we had done in Fig. 5. As for our electrophysiology data, we found that Gaboxadol suppressed the effect of dFB sleep, with consequent daytime sleep intensity being the same as controls (and Gaboxadol alone) for the combined sleep manipulation (Fig. 6e; Supplementary Fig. 8c). This suggests that prolonged dFB activation results in the need for deeper sleep afterwards, so it is unlikely to be achieving exactly the same sleep functions as Gaboxadol-induced sleep during the same period. Interestingly, dFB-activated flies slept almost as deeply afterwards as sleep-deprived flies (Fig. 6e), suggesting a homeostatic rebound to recover lost sleep functions.

To determine possible functional effects of either sleep manipulations, we measured flies' responsiveness to mechanical stimuli while they were awake, after having been 'forced' to sleep for 12 h by either method. Sleep deprivation significantly impairs wakeful responsiveness to the vibration stimuli, with fewer than half of awake flies responding compared to controls (Fig. 6f), and essentially no increase in locomotion speed (Supplementary Fig. 8d). In contrast, wakeful responsiveness was not different from controls following either of the sleep manipulations,

**Fig. 6** Behavioral effects and consequences of dFB sleep vs. Gaboxadol sleep. **a** Flies in glass tubes were filmed from above for the duration of the experiment. DART software was used to track fly activity and test behavioral responsiveness using a mechanical vibration. Red LEDs were used for optogenetic activation (see Methods section). **b** Behavioral responsiveness either probing for general responsiveness or arousal thresholds (top right panels) was tested by quantifying the change in fly locomotor activity following the vibration stimulus (measured in g, see Methods section). Following stimulus delivery (dashed red line), flies increase their locomotion speed as shown by their displacement in the tube (top left panel). Responsiveness could be binned by prior immobility groups, where > 5 min of immobility was considered as sleep (bottom panels). **c** Average sleep duration (± s.e.m.) for flies induced to sleep for 24 h by either optogenetic dFB activation (dFB) or Gaboxadol (GAB) compared to controls. All flies were 23E10-Gal4/UAS-Chrimson and exposed to red light, but dFB flies were fed food containing retinal, GAB flies were fed food containing Gaboxadol, while control flies were fed unadulterated food ($n = 102$ flies for each group). **d** Arousal thresholds (AT) for 23E10-Gal4/UAS-Chrimson flies exposed to the same conditions as in **c**. ***$p < 0.001$, ****$p < 0.0001$, Kruskal–Wallis test with Dunn's multiple comparisons test. $n = 102$ for all groups. Medians (yellow bars) and 75th percentiles (box) and outliers (whiskers) are shown. **e** Daytime behavioral responsiveness of sleeping flies during recovery following 12 h of nighttime sleep induction or sleep deprivation (SD). Sleep induction methods are as in **c** and **d**, or both methods combined (GAB/dFB). *$p < 0.05$, **$p < 0.01$, by ANOVA with Tukey's multiple comparisons. Controls, $n = 153$; dFB, $n = 153$; GAB, $n = 150$; GAB/dFB, $n = 102$; SD, $n = 168$. **f** Daytime behavioral responsiveness of awake flies (see Methods section) during recovery following 12 h of nighttime sleep induction. ****$p < 0.0001$, by ANOVA with Tukey's multiple comparisons. The Data are from the same flies as in **e**. Images: Michael Troup

separately or combined (Fig. 6f). This suggests that either sleep manipulation (and both combined) are accomplishing at least one key function linked to maintaining normal wakeful levels of behavioral responsiveness to the mechanical stimuli. The surprising result here is that prolonged dFB sleep increased the need for deeper sleep afterwards (like sleep deprivation, Fig. 6e), but without compromising wakeful responsiveness (unlike sleep deprivation, Fig. 6f). A homeostatic deeper sleep rebound suggests that some other sleep functions have not been satisfied following prolonged dFB activation. Together with our electrophysiology results, these behavioral data support our overall conclusion that acute dFB activation engages a distinct sleep stage in the fly brain. However, comparisons with Gaboxadol-induced sleep remain speculative because of the different approaches used (neural activation vs. drug intervention).

## Discussion

Sleep in invertebrates has traditionally been studied using behavioral criteria, but the most insight about sleep in vertebrates has come from monitoring brain activity, by measuring electroencephalograms (EEGs) for example. Without easy access to traditional measures of brain activity such as EEG, sleep in invertebrates has tended to be viewed as a single phenomenon, perhaps under the assumption that sleep should be simpler in these smaller animals, compared to mammals and birds for example. Indeed, the few sleep recordings that have been done, mostly in flies and bees, showed that unlike mammals, brain activity levels appeared to simply decrease during sleep in invertebrates[32]. However, a few behavioral studies have identified micro-behaviors during sleep in bees[33], as well as changing arousal thresholds during sleep in flies[11], suggesting that brain activity in insects might be dynamically partitioned by distinct stages. Recent work on sleep in reptiles[6] suggests that partitioning sleep into different stages with potentially different functions is likely to be an ancient feature of sleep throughout evolution. Consistent with our previous behavioral work[11], we identify in this study neural correlates for distinct sleep stages in flies. Transitions in and out of sleep are associated with increased oscillatory activity, and these seem to be governed by the 'sleep switch'[10, 15] in the dFB of the central complex. Interestingly, our results show that dFB-associated sleep does not appear to be any 'lighter' than other forms of sleep in flies. This suggests that lighter and deeper sleep in flies[11] could instead be correlated to overall LFP amplitude (1–100 Hz), while the dFB-associated oscillations may represent a distinct sleep stage.

Different forms of oscillatory brain activity have been used to identify sleep stages in mammals[34], although whether these oscillations accomplish any sleep functions remains debated[35]. Slow wave or 'delta' sleep (1–4 Hz) has been implicated in synaptic homeostasis[36], spindles (12–14 Hz) are thought to inhibit responsiveness[37], and sharp wave ripples (140–200 Hz) have been associated with memory replay[38]. During wake, alpha waves (7–11 Hz) have been associated with drowsiness and perceptual inhibition[35, 39]. In comparison, the invertebrate brain does not display as rich a repertoire of strong oscillatory activity, although there is evidence of local field potential oscillations associated with visual and olfactory processing in insects[40, 41]. The 7–10 Hz oscillations we have identified during sleep transitions in the fly brain could be accomplishing a similar function to some sleep-related oscillations in the mammalian brain. For example, it is possible that the 7–10 Hz oscillations serve a similar role as has been proposed for sleep spindles during stage 2, for blocking the processing of external stimuli[37, 42].

To our knowledge, the only other evidence of rhythmic brain activity during sleep in invertebrates is in crayfish[43], where oscillations in an adjacent frequency range (15–20 Hz) have been described and further characterized[44], and these also appear to be generated in the central complex of these arthropods[45]. Together, our fly data and the crayfish sleep studies suggest that oscillatory brain activity may be a common feature of sleep across the wide range of invertebrates that share a similar brain architecture featuring a central complex[46]. Our findings show that the sleep-related oscillations in the fly brain predominate during the beginning and end of spontaneous sleep bouts (Fig. 1g), suggesting a timeframe for this sleep stage. Our localization of these oscillations to the dFB suggests the neuroanatomy likely to be involved in generating these sleep-related oscillations, although it remains unclear why artificial activation of the dFB (as seen in our optogenetic experiments) produces a broader range of frequency effects. Since the dFB has previously been implicated with a sleep switch or homeostat[15], this nevertheless suggests that the sleep switch promotes a distinct 'oscillatory' sleep stage before other forms of sleep take over. This view would still be consistent with the general consensus that, on average, sleep is associated with decreased LFP activity in flies[11, 19, 20] and other invertebrates[24–26]. Also consistent with this view, we found that the main effect of the sleep-promoting drug Gaboxadol[18] was to decrease LFP amplitudes in the fly brain.

If fly sleep is primarily characterized by decreased brain activity, then why does the fly 'sleep switch' produce increased oscillatory activity upon sleep induction? One possibility could be that the dFB plays a larger role than just promoting sleep. Central complex neurons, including those projecting to the dFB, are probably engaged in modulating sensory information processing more generally, in awake animals as well[47, 48]. Some central complex circuits, including the dFB, could be required for attention-like processes for example[49], which would involve selective suppression of sensory stimuli, or at least a form of gain control. Synchronized activity from dFB neurons, as we have found here during sleep, might effectively interfere with ongoing wake-related dFB processes, as a first step to turning off attention and falling asleep. To test this idea, that sleep and wake processes might be related at some level[50], will require a better understanding of how oscillatory brain activity might be deployed differently during wake and sleep to modulate behavior. Future research in *Drosophila* should reveal whether the fly brain uses this strategy to regulate behavioral responsiveness.

## Methods

**Animals**. Flies (*Drosophila melanogaster*) were reared on standard yeast-based *Drosophila* medium under a 12-h light and 12-h dark cycle (lights on at 8 A.M.). Three experimental setups were used: overnight brain recording setup (Fig. 1a), multichannel brain recording setup (Fig. 2a), and exposed-brain recording setup (Fig. 3a). Flies used for overnight brain recording experiments were kept in the same room to allow exposure to the same daily fluctuations in temperature (22–24 °C) and humidity (40–60%) as during the experiments. All other flies were raised at 25 °C with 50–60% humidity. Adult female flies (<7 days post-eclosion) were used for all experiments. Wild-type Canton-S (CS) flies were used for overnight recording experiments. UAS-TrpA1 and UAS-2xEGFP were acquired from the Bloomington *Drosophila* stock center. The Gal4 drivers used for driving expression in the dFB neurons were C5-Gal4, 104y-Gal4 and 23E10-Gal4, also from the Bloomington *Drosophila* stock center. UAS-CsChrimson was kindly provided by Vivek Jarayaman (Janelia Research Campus).

For all experiments, flies were anesthetized on a thermoelectric-cooled block (1–2 °C). To prepare the fly for both the overnight and multichannel recording experiments, the dorsal surfaces of the fly head and thorax were secured to a tungsten rod[11, 21] using dental cement (Coltene Whaledent Synergy D6 Flow A3.5/B3) and cured by 30–40 s exposure to high intensity blue light (Radii Plus, Henry Scheinn Dental).

**Two channel differential LFP**. As described previously[11], to perform the overnight recordings (Fig. 1a), we used pulled borosilicate micropipettes (World Precision Instruments TW100F-4, pulled using a Sutter P-97 micropipette puller), which were cut, leaving only the 6 mm length of the tip (~3MΩ resistance), and subsequently filled with extracellular fluid (ECF) containing (in mM): 103 NaCl, 10.5

trehalose, 10 glucose, 26 NaHCO$_3$, 5 C$_6$H$_{15}$NO$_6$S, 5 MgCl$_2$ (hexa-hydrate), 2 sucrose, 3 KCl, 1.5 CaCl (dihydrate), and 1 NaH$_2$PO$_4$. The cut micropipettes were then carefully inserted ~100 μm into each brain hemisphere through the dorsal eye rim using a mechanical micromanipulator, with each micropipette permanently held in place using dental cement. Fine tungsten wire electrodes (25 μm; A-M Systems) were inserted into the solution-filled micropipettes and sealed within the micropipette using electrical insulating compound (Dow Corning 4). The prepared fly was then placed onto an air-supported polystyrene foam ball that served as a walking/resting platform (Fig. 1a). Local field potentials (LFPs) were recorded, 1–2 h after implanting the micropipette electrodes[11], using field-effect transistors (FETs) (NB Labs, Denison, TX). Recordings were performed at a sampling rate of 291 Hz and amplified (×10,000 gain) with a differential amplifier, signal bandpass filtered (low: 1 Hz, high: 100 Hz) (Warner Instruments DP-304), digitized (National Instruments BNC-2090), and the data acquired with a custom-built software on a LabVIEW platform[11]. The electrophysiology setup was housed within a light-shielded box to allow a controlled environment of 12-h light and 12-h dark cycle. Infrared LEDs illuminated the fly for movement monitoring via an infrared-enabled webcam (Logitech Pro 9000, with modification described below), producing monochromatic low-resolution images (27 × 34 pixels) with a frame rate of 3 frames per second, well-suited for a continuous long recording session. Movement data were quantified offline using a custom script in MATLAB (The Mathworks, Natick, MA) and subsequently time-matched with the LFP data.

Most readily available webcams have an infrared filter, which was removed in order to film under infrared lighting conditions. This first involved accessing the camera's circuit board by unscrewing the outer case, then removing the screws holding the lens in place, followed by de-soldering the 2 connectors between the lens assembly (auto focus unit) and the circuit board, to allow access to the rear of the lens. The thin glass disc (the IR filter) was removed by breaking the glass with a pair of forceps, ensuring that none of the glass pieces fell into the photo sensor underneath. Once the IR filter was removed, the webcam was reassembled to its original state. A visible light filter was fitted to the front to complete the modification.

**Multichannel LFP**. Methods for performing multichannel fly brain recording have been described previously[21]. Briefly, to record from multiple channels in the fly brain we used a 16-electrode linear silicon probe (model no. A1 × 16-3 mm50-177; NeuroNexus Technologies). The probe was inserted into the flies' eyes laterally, perpendicular to the curvature of the eye, with the aid of a micromanipulator (Merzhauser, Wetzlar, Germany) (Fig. 2a, middle panel). We inserted the probe such that the electrode sites faced posteriorly within the brain. A sharpened, fine tungsten wire (0.25 mm; A-M Systems) served as a reference electrode and placed superficially in the thorax. Recordings were made using the Tucker–Davis Technologies (Tucker-Davis Technologies, US) multichannel data acquisition system at 25 kHz coupled with a RZ5 Bioamp processor and RP2.1 enhanced real-time processor.

**Exposed-brain targeted single channel LFP**. For experiments on the exposed-brain assay (Fig. 3a), the two forelegs were cut in the femur segment and the proboscis restrained with dental cement to the ventral thorax. This was done to provide access to the posterior surface of the head and to eliminate proboscis or foreleg movement from disrupting the brain visualization and electrical recording. The flies were then mounted and sealed with dental cement onto a custom fly plate[51, 52] that provided electrode access to the posterior head (Fig. 3b). The bath chamber of the fly plate was filled with oxygenated ECF (95% O$_2$, 5% CO$_2$), immersing the brain, while keeping the rest of the fly dry. With the use of a pair of forceps and 30½ gauge syringe needle, the head was dissected, with the perineural sheath removed either mechanically with forceps or chemically using protease (0.5% collaganase type IV solution). Similar to the overnight setup, the fly in this preparation was also positioned on an air-supported ball. The fly brain was kept healthy with a continuous delivery of oxygenated ECF at a flow rate of about 2 ml/min. LFP recordings were performed with a glass electrode amplified (via FETs) and filtered (low: 0.1 Hz, high: 1 kHz) (A-M Systems Model 1700), digitized (Axon Digidata 1440 A Digitizer) and sampled at 1000 Hz using the data acquisition software AxoGraph × 1.4.4 (Axon Instrument) on a computer running Windows XP. A fixed-stage upright fluorescence microscope (Olympus BX51WI, U-RFL-T, Olympus, Berlin, Germany) was used to visualize the fly brain, and a motorized micromanipulator system (Sutter MP-285) was used for guiding electrode insertion. The fly was illuminated using a 3 mm white LED (PN: 5219901802 F, Dialight, South Farmingdale, NJ) placed at a distance of 6–8 cm from the fly for behavioral monitoring using a camera (Point Grey GRAS-14S3C-C) at a resolution of 480 × 640 pixels and 30 frames per second. For optogenetic experiments, illumination of the fly was achieved using an infrared LED (Osram SFH 4232) instead of a white LED, coupled with a custom lens filter fitted to the camera that specifically blocks out red light. Behavioral data were acquired and stored on a computer running Linux OS. For optimal visualization of the targeted neurons, a second high-powered infrared LED (Osram SFH 4232) was used with its light path redirected to the fly's right eye via an optic fiber (1 mm diameter), positioned about 2–3 mm from the eye (Fig. 3a). A microscope camera (DAGE-MTI IR-1000) connected to an LCD TV unit (Samsung SyncMaster 940MG) provided live imaging of the fly brain and neurons. Visualization of the GFP-labeled neurons was achieved using a

mercury short-arc lamp (HBO 103 W/2). No GFP-labeling was used to target specific sites in the optic lobes recording and were therefore only approximated. The recording site was confirmed by releasing dye in a subset of flies (Fig. 3c, and see immunolabeling, below).

**Arousal-testing stimulus for tethered flies**. Methods describing the use of a vibration stimulus for testing behavioral responsiveness of tethered flies in the overnight recording setup was previously described[11]. Briefly, a vibration stimulus generated by a 12 mm shaft-less vibrating motor (Pico Vibe 312-101; Precision Microdrives) was delivered to a subset of flies in the overnight recording pre-paration. We then examined the flies' behavioral responsiveness, from the movie images, to determine whether flies in the brain-recording setup were sleeping as defined by an increased arousal threshold. The motor was glued to the top end of the brass tether rod (Fig. 1a), delivering a vibratory stimulus of 1 V intensity to the fly through the length of the rod lasting < 1 s at 15 min intervals throughout the recording session. Stimulus delivery was automated and set using a custom MATLAB script[11].

**Thermogenetic and optogenetic sleep induction**. Thermogenetic sleep induction in the multichannel brain recording setup was achieved by heating the suspended fly from below, using a 100-W halogen lamp (Zeiss) equipped with an infrared long pass filter[21]. For the exposed-brain recording setup, the fly brain was heated directly by modulating the temperature of the ECF bath solution. This was achieved by using an in-line heater/cooler (Warner Instruments Model SC-20), driven by a temperature controller (Warner Instruments Model CL-100), and cooled by a liquid cooling system (Warner Instruments Model LCS-1). With the aid of a thermistor, the temperature of the bath was kept at room temperature in the range of 22–23 °C. During the stimulation period, temperature was ramped up to >29 °C after 2 min of room temperature recording (baseline), and lasted for 5 min before returning to <23 °C for 5 min of recovery (see Fig. 3d, bottom plot). Temperature throughout each experiment was handled by AxoGraph.

For optogenetic experiments, dietary supplements of ATR were needed for the transgenic channelrhodopsin to function. Therefore, all flies used for optogenetic experiments were transferred to food vials containing 1 mM ATR supplementation[53] at least 2 days prior to experimentation. The activation stimulus consisted an ultra-bright red LED (617 nm Luxeon Rebel LED, Luxeon Star LEDs, Ontario, Canada) directed to the opened section of the fly head (Fig. 5a, bottom panel), producing 0.1–0.2 mW/mm2 at a distance of 4–5 cm with the aid of concentrator optics (Polymer Optics 6° 15 mm Circular Beam Optic, Luxeon Star LEDs). To prevent overheating the fly and the immediate environment, the LED was mounted onto a sink pad (SinkPAD-II 20 mm Star Base), which was attached to a small heat sink. The temperature of the solution bath was also kept constant by the temperature controller system (see above). Continuous light exposure was administered after 1 min of baseline recording and lasted for 2 min (Fig. 5b). Timing of the light switch was controlled by AxoGraph, which also measured the timing of light exposure from a photodiode (Fig. 5a, bottom panel).

**Pharmacologically induced sleep**. The GABA$_A$ agonist, Gaboxadol, also known as 4,5,6,7-tetrahydroisoxazolopyridin-3-ol (THIP), was used to induce sleep in flies[16]. Instead of feeding, as in previous studies[16, 28], Gaboxadol was delivered directly to the fly brain by dissolving it into the ECF[28]. Three concentrations were used (in mg/ml): 0.05, 0.1, and 0.2. The Gaboxadol-containing ECF was delivered to the bath chamber at the rate of 2 ml/min for a total of 5 min, after 5 min of recording with standard ECF, and immediately washed out by switching back to standard ECF thereafter. The drug delivery setup consisted of two 50 ml reservoirs, one with Gaboxadol-containing solution and the other standard ECF, both connected to a 3-way solenoid valve with the outlet leading to the fly plate bath chamber (Fig. 4a). The timing for the switching of the solenoid valve was controlled by AxoGraph. The effect of optogenetic activation on Gaboxadol-induced sleep flies was examined by first running the optogenetic activation protocol (see above, with a baseline recording of 5 instead of 2 min), followed by a 5-min delivery of Gaboxadol solution, and subsequently running the optogenetic activation protocol for the second time (5 min baseline, 5 min activation, and 10 min of recovery; see Fig. 5d, top panel).

**Immunolabeling**. The electrode positions in the fly brain were labeled with Texas Red fluorescent dye (Invitrogen) via iontophoresis to confirm the recording loca-tion in the dFB (Fig. 3c). Fly brains were dissected and fixed in 4% paraf-ormaldehyde in a phosphate buffer solution (PBS). After a minimum of 1 h in fixative, the brains were washed with 0.2% Triton X-100 in PBS (PBST) with 0.01% sodium azide (Sigma), blocked in 5% normal goat serum in PBST, and let incubate overnight in a primary antibody solution (1:10 mouse anti-nc82 + 1:1000 rabbit anti-GFP + block solution). The next day, the brains were washed in PBST and let incubate overnight in a secondary antibody solution (1:250 goat anti-rabbit Alexa Fluor 488 and 1:250 goat anti-mouse Alexa Fluor 633). The brains were washed in PBST for the final time and embedded in Vectashield and imaged using a confocal microscope (Zeiss).

**Behavioral analyses of tethered flies**. Movie images of the flies acquired from the overnight and exposed-brain recordings were analyzed and quantified in MATLAB using a pixel subtraction method[11], generating the pixel change value (Δ pixels), which quantifies the fly's behavioral activity. Image noise level varies with each movie recording and was therefore determined for each recording by visually inspecting the activity trace and assigning a threshold value. The fly was considered active during the times when the measured activity exceeded this threshold[11].

For each stimulus trial in the arousal-testing experiments (see Arousal-testing stimulus for tethered flies), the average Δ pixels in the 15 s post-stimulus were calculated, and if exceeded the threshold (see above), the fly was regarded to respond to the stimulus (respond group), while for trials with values below threshold, regarded unresponsive (did not respond group; Fig. 1b, top). Response rate was thus calculated as the averaged percentage of trials when the flies responded (Fig. 1b, bottom). Visual inspection on an overnight fly movie recording revealed a range of non-locomotion micro-behaviors, which we classified into one of three groups: posterior groom, anterior groom, and proboscis extension. Times of occurrence for each of the micro-behavior in one fly recording were determined manually, and subsequently time matched to the LFP recording (Supplementary Fig. 2). Behavioral activity was not monitored for flies in the multichannel recording setup.

For comparing fly activity in the exposed-brain recording setup, Δ pixels were reduced to a binary format such that behavioral activity was quantified as the percentage of frames where Δ pixels exceeded the threshold in a specified time range (Figs 3f and 4c). For Gaboxadol-induced sleep experiments, we observed a rapid decline in behavioral activity following drug perfusion, which we defined as the onset of the drug's effect. We observed some variability in the latency period of the drug effect onset across flies, and therefore the drug onset time was determined for each fly by examining the movie recordings. Comparison between the percentage movement in the period within 5 min prior and 5 min after drug onset were made to confirm the cessation of movement that occured as a result of Gaboxadol exposure (Fig. 4c–e). Latency periods were defined as the time it takes since the commencement of drug perfusion to the onset of behavioral effect of the drug.

**Overnight recordings**. Analyses on the LFP data obtained from the overnight recording setup were performed offline on custom scripts in MATLAB (2014a, 2015a). Analyses were restricted to frequencies between 0 and 100 Hz as activity above 100 Hz in the fly brain is unlikely biological. For comparing the LFP activity across different arousal states sorted into day and night (Fig. 1d), the raw LFP were split and grouped based on the recorded movement data (see Behavioral analyses of tethered flies) and time-of-day during the recordings. The raw LFP for each condition were then transformed into power using the Morlet wavelet transformation function "ft_specest_wavelet" in the Fieldtrip MATLAB toolbox[54]. The width setting of the wavelet used was set at 30 with 3 standard deviation (gwidth). Power differ in magnitudes across fly recordings, and were therefore normalized for each fly prior to averaging. Normalization involved obtaining the mean values for the power in the Wake day condition, and used as the reference (denominator of a ratio calculation) to compare with the individual power values of the other 3 conditions (numerator). The resulting values used for statistical analyses were therefore ratio values of power in each group relative to those for Wake day. For the 0–100 Hz analysis, this normalization process was performed separately in binned groups of 2 Hz prior to averaging.

Similarly, for sleep bout LFP analysis, the mid-sleep section was used as the reference with ratio of power in each sleep segment within a sleep bout obtained prior to averaging the ratio values across all sleep bouts (separated into day and night) in a fly, and subsequently averaged across all flies. This process of normalization was applied in the 7–10 Hz analysis (see Fig. 1g), where the ratio calculation was performed first in binned groups of 0.1 Hz for each sleep bout prior to subsequent averaging. Additionally, we examined the LFP power in a series of broader frequency domains (Supplementary Fig. 3a–e), identified previously in a k-means cluster analysis of Drosophila brain activity (2–6 Hz, 6–15 Hz, 15–30 Hz, 30–50 Hz, 50–100 Hz)[21] excluding 0–2 Hz due to potential heartbeat artifacts.

Power spectra were generated by performing discrete Fourier transform on the raw LFP data (fft function from MATLAB Signal Processing Toolbox) (Supplementary Fig. 1b). To prepare the time-frequency spectrograms, the data were first lowpass filtered at a cutoff of 100 Hz and then highpass filtered at a cutoff of 0.2 Hz by using a second-order Butterworth filter, with further processing (tapers [3 5], moving window [1 0.05]). The time-frequency spectrogram was generated by the mtspecgramc function in the Chronux MATLAB toolbox[55]. For the analysis of the frequency domain, Letswave, Letswave 5 (http://nocions.webnode.com/letswave) was used, which runs on MATLAB 2015a. As before, the data were first lowpass filtered at a cutoff of 100 Hz and the highpass filtered at a cutoff of 0.2 Hz by using a second-order Butterworth filter. Furthermore, the data were cropped and divided into 4 categories (day wake, day sleep, night wake, night sleep). The Data were first averaged in the time domain for each category and each animal, then a discrete Fourier transform was performed on the averaged data and the data were normalized. The signal to noise ratio (SNR) was calculated as the ratio between the amplitude for each frequency and the mean amplitude of 15 neighboring frequency bins (0.1 Hz) on each side. Z-scores for the frequency peaks

were calculated in a similar way as the SNR. Z-score values above 1.64 indicated a significant ($p < 0.05$) difference between peak and baseline.

For determining whether sleep-related oscillations were homeostatically regulated in overnight experiments, we divided all nighttime sleep bouts (>5 min) into three equal-sized epochs per fly: early sleep, mid sleep, and late sleep. Normalized LFP power for the 7–10 Hz domain of the first night epoch was compared to the last epoch, and any differences were tested by a Wilcoxon matched-pairs signed rank two-tailed test. We further analyzed whether 7–10 Hz oscillations during spontaneous sleep is associated with increased responsiveness following a vibration stimulus. For that purpose, we performed a Morlet wavelet transformation (2–15 Hz), as described earlier. We normalized all the data ([0 1]) for every fly separately and extracted the average sleep LFP power for the 7–10 Hz frequency range. We then separated our data into low 7–10 Hz LFP power and high 7–10 Hz LFP power based on a set threshold defined by the average LFP power of the neighboring frequencies (2–7 Hz and 10–15 Hz). Vibration stimuli occurred every 15 min throughout the night, as described above. All vibration stimuli that coincided with 'high' 7–10 Hz LFP power during a sleep epoch were noted, and a behavioral response rate was calculated as before (see Behavioral analyses of tethered flies). Response rates were compared for trials when the stimulus coincided with 'low' 7–10 Hz LFP power during sleep epoch. For all trials, 7–10 Hz LFP power was determined for the 10 s preceding the vibration stimulus.

**Multichannel recordings**. All LFP data were analyzed offline in MATLAB (2015a). Raw LFP data were down sampled to 1000 Hz, filtered between 0.5 Hz and 200 Hz using a fourth-order Butterworth filter. Bipolar-referencing to the most lateral channel (1, in the optic lobe) was used to create 15 differentiated channels. From this, independent components analysis (ICA) was conducted to reduce spontaneous artifacts in the data using the FastICA function[56, 57]. From the data set, 3 s epochs were extracted for each condition prior to the heating condition as baseline, the 'Heat ON' condition, as well as a post heat condition.

Drosophila heart beat has been shown to change frequency during heating[58] which could be a confounding factor in our experiments. Thus, channels containing clear heartbeat artifacts, at any stage of the experiment (baseline, heat on, heat off) were removed from subsequent analysis. For this reason, a multi-tapered Fourier transform was performed using the mtspectrumc.m Chronux function[55] to improve resolution in order to visually identify channels contaminated with a heartbeat around 2–4 Hz and its harmonics[21].

We converted LFP into power as described above (see Overnight recordings). For this, we used a wavelet resolution of 3 s corresponding to the length of each epoch, and a wavelet width of 3 s.d. This was done twice, once to look at the 2–40 Hz frequency band across channels, and again to examine differences in the 5 frequency bands described above (2–6 Hz, 6–15 Hz, 15–30 Hz, 30–50 Hz, 50–100 Hz; see Supplementary Note 1 for analysis).

For normalization of the power values, we divided each channel by the median of the baseline activity, followed by the median by channel groups for every fly. The bipolar-referencing scheme allowed the orthogonal selection of channels by grouping them in 3 groups of up to 5 channels (optic lobe 1, center, optic lobe 2). The resulting data were organized in factor coded columns and exported to R version 3.3.2[59] for further statistical analysis.

A non-parametric multi-factor ANOVA was used to assess statistical significance on the mean LFP power, with post hoc contrasts on a three-way interaction term between brain regions, fly line, and heat condition. Significant effects were determined at a Bonferroni corrected alpha value of 0.0125.

**Exposed-brain recordings**. All analyses on the LFP data obtained from the exposed-brain recording setup were performed offline on custom scripts in MATLAB (2014a, 2015a). The time-frequency spectrograms were generated in the same way as described for the overnight recordings. To obtain the averaged spectrogram across multiple flies (Figs 3e, i and 5b, bottom), the data were first normalized for every animal by dividing the amplitudes of frequencies over time by the mean amplitude of the baseline for all frequencies. Then, the ratio was calculated by dividing all values by the maximum amplitude of the baseline. Subsequently, data of all experimental animals was averaged.

For quantifying the LFP signal, wavelet transformation was first applied to the raw LFP data prior to splitting into 3 groups: baseline, stimulus on, and stimulus off. The power values for the stimulus on and stimulus off groups were divided by the mean of the baseline power within the corresponding frequency bins (same frequency domains and bins as the analysis for the overnight recordings). Additionally, the ratio power values were zeroed such that any negative power value indicates a decrease of LFP activity in relation to baseline. For thermogenetic experiments, segments of data where the temperature transitions between the two stable states (24 °C and 29 °C) were excluded from analysis. LFP recordings from the optogenetic experiments contain obvious artifacts during the brief period when the light was switched on and off; therefore, a short data segment (from 5 s prior to 5 s after the light switching) were excluded from analysis. To examine the LFP effect of Gaboxadol-induced sleep, we compared the LFP power between the 5 min prior and the 5 min after the drug effect onset (see Behavioral analyses). Likewise, the power used for this comparison were first normalized to baseline values which was the first 5 min of each recording.

Similar to the multichannel brain recording, we detected oscillatory activity surrounding 2 Hz that likely originated from the heartbeat. The intensity of the heartbeat can often be observed visually under the light microscope during cuticle dissection. We first attempted to stop these muscle contractions by mechanically damaging the relevant muscle[58] with a pair of forceps. We then also excluded any observed LFP effect in the 0–2 Hz domain as it is likely contaminated by the movement artifacts. In some flies, however, the harmonics of the heartbeat artifact were also present, clouding any signal that manifests in frequencies above 2 Hz. We excluded these recordings entirely, based on the interpretation of heartbeat artifacts by two experimenters (M.H.W.Y. and M.J.G.) independently.

**Statistical analyses**. All statistical analyses for data gathered from the overnight and exposed-brain recording setup were performed using Prism 7 for Windows (GraphPad). A subset of behavioral and LFP power data set did not pass the Shapiro–Wilk normality test ($p < 0.05$). Depending on the outcome of the Shapiro–Wilk normality test, a Wilcoxon signed rank test or a $t$ test was used to test for significant effects between two matched conditions. The appropriate tests used are mentioned in the figure legends. Friedman test with Dunn's post hoc multiple comparisons test were used to compare three or more matched conditions, and Kruskal–Wallis test with Dunn's post hoc for unmatched data. All the data presented in figures are as means ± S.E.M. for bar and line graphs while box and whiskers plot presents median and 10–90 percentiles as whiskers. All tests for significance were two-tailed and confidence levels set at $\alpha = 0.05$.

For the multichannel statistical analysis, the following R packages were used: ARTool[60, 61], car[62], dplyr[63], influence.ME[64], lattice[65], lme4[66], magrittr[67], MASS[68], Matrix[69], nortest[70], phia[71], and plyr[72]. The data.frame was organized by splitting the data set into 104 y and C5 groups to be analyzed separately. In the case of the frequency cluster analysis, the data were further divided into individual frequency bands. The data for the 2–40 Hz band were not normally distributed (Lilliefors (Kolomogorv–Smirnov) Test $p < 0.001$). Therefore, a non-parametric test was used for the log transformed data, which allowed the test of multiple factors and their interactions called the Aligned-Rank ANOVA from the R ARTool package[61]. The Aligned-Rank ANOVA allows multi-factor or mixed model regression to be performed on a non-parametric dataset or one that violates the normal assumptions of parametric models[61]. For the 2–40 Hz and frequency cluster analysis, Aligned-Ranks were constructed using the art function from ARTool. The ARTool package makes use of the lmer function for testing mixed models from the lme4 package and thus uses its syntax.

To perform contrasts on significant higher-order interactions, the testInteractions function from the phia package was used to test post hoc contrasts between categorical variables, employing a scheme called Helmert coding[73]. Unlike other types of factor level coding, Helmert contrasts allows flexibility in the equivalence assigned to factor levels[73]. In this instance, it allows the mean across both optic lobes to be compared to the center for the Region factor (e.g., −1/2 for each optic lobe and 1 for the center, summing to zero). The contrasts also compared the TRP-lines to GAL4 or UAS controls (TRP = 1, GAL4 = −1), Heat On to Baseline (Baseline = −1, Heat On = 1, Heat Off = 0) or Heat Off to Baseline (Baseline = −1, Heat On = 0, Heat Off = 1), unless otherwise specified. The testInteractions function takes the model output provided by ARTool. The Aligned-Rank ANOVA has two diagnostic tests associated with it which tests whether the aligned-rank transformation was performed successfully[61]. For the first test, the columns of aligned-rank responses should all sum to zero. All analyses performed passed this test. The second test checks whether a full-factorial ANOVA on ranked (but not aligned) responses has all main effects stripped out as indicated by an F value of 0 (Pr = 1).

**Arousal testing in freely walking flies**. Sleep-related metrics (sleep intensity, arousal thresholds, sleep duration) for freely walking flies (Fig. 6a) were determined using the *Drosophila* ARousal Tracking system (DART) as previously described[16, 31]. Twenty-four hours prior to experiments, 3- to 5-day-old adults were collected and loaded individually into 65 mm glass tubes (Trikinetics) that were plugged at one end with standard yeast-based fly food, containing either 0.1 mg/ml Gaboxadol or 0.5 mg/ml ATR. Controls were placed onto normal food and housed under the same conditions as the experimental groups. The tubes were aligned on platforms (6 total platforms, 17 tubes per platform) for filming. Flies were exposed to ultra-bright red LED (617 nm Luxeon Rebel LED, Luxeon Star LEDs, Ontario, Canada) for the duration of the experiment for optogenetic activation of flies fed with ATR. For determining arousal thresholds, flies were probed once every hour for 48 h, with a succession of vibrational stimuli of increasing strength, from 0 to 1.2 g. Each stimulus consisted of 5 pulses of 200 ms, and was delivered in 0.2 g increments 15 s apart. To investigate behavioral responsiveness, flies were stimulated every hour with 5 succesive vibrations of equal strength (1.2 g), 200 ms apart. Sleep intensity was measured as the proportion of immobile (sleeping, as per >5 min criteria) flies that responded (at any level) to these stimuli. Flies were determined to have responded if they moved by a threshold of at least 3 mm (~3 body lengths) within the minute following the stimulus, as reported previously[16, 31]. To determine awake responsiveness, we excluded sleeping flies (i.e., flies immobile for five minutes or greater prior to the stimulus) and only flies that had moved within the four minutes prior to the stimulus (*i.e.* awake flies) were included in the analysis. Awake responsiveness was measured as the proportion of awake flies responding (Fig. 6f),

as well as their response magnitude (Supplementary Fig. 8c). To determine response magnitude, fly activity was averaged for two minutes prior to and 15 min after each stimulus. This average activity was fitted with a single-inactivation exponential equation and the peak amplitude of activity following the stimulus was measured. For experiments testing the effect of different sleep induction methods on subsequent behavior, flies were placed on either 0.1 mg/ml Gaboxadol, 0.5 mg/ml ATR, or drug-free food in vials for 12 h (8 P.M.–8 A.M.) while exposed to red light, and then transferred to DART for 12 h (8 A.M.–8 P.M.) for arousal probing. Sleep deprivation was performed using SNAP devices as described previously[16, 31].

**Code availability**. The code used to generate the results that are reported in this study are available from the corresponding author upon reasonable request.

**Data availability**. The data that support the findings of this study are available from the corresponding author upon reasonable request.

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

## Acknowledgements
We thank Leonie Kirszenblat, and Deniz Ertekin for help and comments on the manuscript. We thank Adam Hines for help with behavioral experiments. We thank Richard Faville and Ben Kottler for help with DART. This work was supported by an NIH grant RO1 NS076980-01 to P.J.S. and B.V.S., and an NHMRC grant APP1065713 to B.V.S.

## Author contribution
M.H.W.Y. performed the experiments, analyzed the data, and wrote the paper, M.J.G., R. J., and M.T. analyzed the data, C.R. and A.C.P. performed the experiments and analyzed the data, B.V.A. and P.J.S. helped design the study and contributed reagents, B.V.S. designed the study and wrote the paper.

## Additional information

**Competing interests:** The authors declare no competing financial interests.

