## [Peer Review File · Nature Communications]

Reviewers' comments:

Reviewer #1 (Remarks to the Author):

From behavioral and electrophysiological perspectives, sleep is not a homogenous state, and consists of multiple sub-states, with potentially different functions. This is well-established in mammals but less so in invertebrates, such as fruit flies. The van Swinderen lab previously described different sleep states associated with different levels of arousal threshold in van Alphen et al (2013). Here, they expand upon those findings by performing an array of electrophysiological experiments in vivo. The take-home message of this work is that a 7-10 Hz oscillatory signal seems to identify a transitional stage of sleep, which is seen in flies with natural sleep as well as dFB-activated sleep. In contrast, sleep induced by gaboxadol is not significantly associated with this 7-10 Hz signal. While this manuscript does not necessarily provide major conceptual advances, it provides rigorous data from technically challenging approaches. Because this is largely a descriptive paper, it is important to more fully characterize this 7-10 Hz oscillation associated sleep.

Major comments:

1. In wild-type sleeping flies, the authors should assess arousal thresholds associated with sleep with greater vs reduced 7-10 Hz oscillations. Instead of shaking, the authors could use air puffs to test arousal threshold in this experiment. One might expect that the sleep associated with more 7-10 Hz oscillations to be lighter.
2. I am interested in the macro-architecture for sleep related to this 7-10 Hz signal. The authors show that within a single sleep bout that the 7-10 Hz signal is greater entering and exiting sleep. What about at a 6 or 12 hr level? For example, in mammals, there is greater SWS in the 1st part of the night and more REM in the latter part of the night. Is there generally more sleep with 7-10 Hz signal in the late night compared to the early night?
3. Is 7-10 Hz associated sleep regulated by circadian or homeostatic forces? One may get a sense of this by providing data for the above point. However, to directly address this, the authors should also repeat this in clock mutants and also following 12 hours of sleep deprivation.
4. Overall, I did not find Figure 5 helpful, as I do not think that combining Gaboxadol with dFB activation yields particularly useful information since both drug dose and thermogenetic activation can be varied. I was also confused by the arousal threshold data in Figure 5e. It seems that the arousal threshold is greater when dFB neurons are activated compared to when Gaboxodal is administered, while I was under the impression that Gaboxodal-induced sleep is deeper. Also, there is very little signal range in Figure 5e (particularly at night)—the authors should try stronger stimuli to bring out potential differences in arousal threshold.

Minor comments:

1. In figure 1, when electrodes were inserted into the optic lobes, did the authors confirm electrode placement by post-hoc labeling?
2. Can the authors provide more details on how their Logitech webcam was modified to detect infrared signal?

Reviewer #2 (Remarks to the Author):

The authors' main claim is that they have identified different phases of sleep in *Drosophila*, which seem to 'have different effects on behavior' (line 24). They establish this by recording field potentials in flies whose movements (and thereby sleep state) are simultaneously measured by their rotations of a spherical treadmill, and showing that rhythmic activity in a particular frequency band (7-10Hz) is more prominent at the beginning and end of a sleep bout (Fig1g). This is quite a nice technique, and the use of an electrode targeted specifically to the Fan-shaped Body (a brain area implicated in sleep) in Fig3 is a good addition to the relatively non-directed use of a multi-electrode array.

Overall the paper is a reasonable contribution to the literature. The first part of their claim, the different sleep phases, is reasonably convincing. However, the second part, what role different phases play behaviorally doesn't have a strong answer: the authors just show that recovery sleep is different after two different ways of inducing sleep (Fig 5f). What that accomplishes is not addressed. But the first part is still a valuable contribution, and I think the second part is largely a matter of changing the language to tone down that aspect of their claims. If the authors could change the language to avoid the implication that they have shown these phases of sleep are 'functionally distinct' (e.g. in the Abstract, changing the section heading 'Behavioral Effects...' to Behavioral correlates), and generally making it clear that their evidence stops at behavioral correlates, the statements would be more justified.

I have two major critiques:

1) The experiment showing an increase in 7-10hz power with dFB-induced sleep needs larger n (Fig 2b and Fig 3g). In Fig2b, for the Trp/104y flies, I cannot see why the only significant effect observed is the middle red bar, and not also the middle blue bar (post-heat), based on the magnitudes and the error bars. If anything the magnitude of the increase in power is larger post-heat than heat-on. I think the authors need to dig into the statistics here and see what's the basis for the significance. They should also increase the n since there's absolutely no reason why channels 1-5 and 11-15 are so different from one another in the Trp/104y panel (compare top and bottom blue bars; also true but smaller effect comparing top and bottom red bars). Clearly these differences must come from noise. Given that those noise-based differences are quite large, I think they need much more than n=7 to reliably detect the effect they are interested in.

The FB-targeted recordings in Fig 3 are a better technical approach than the trans-brain recordings in Fig 2. But again in Fig3g, the Trp/104y flies 'Heat On' show a very small effect, but what is noteworthy is that the Post Heat bars for these flies show very low variability, lower than the +/104y bar, and lower than the OL recordings in Fig3j. The fact these control/no effect conditions fail to show a consistent result again suggests that they haven't yet got a large enough sample size.

The critical piece of information showing that there are different states during normal sleep is the plot in Fig 1g, indicating power in the 7-10hz band fluctuates. It is important the authors show a plot similar to Fig1g for GAB-induced sleep to be able to make a direct comparison. Clearly there are still some 7-10hz bouts in their data with GAB-sleep (they point them out in Fig4g). They never directly test whether the those bouts are still more prominent at the beginning & end of sleep epochs in the GAB-induced sleep.

Finally, I suggest moving the experiments with the other dFB driver in Sup Fig 4 to the main Fig 2. 104y clearly labels many non-dFB neurons, so having a partly overlapping driver in c5 is very important to increase confidence the effect is really derived from the FB expression.

2) With the LFP recordings, when they combine GAB- and dFB-induction of sleep, the oscillations they

observe from dFB go away (Sup Fig 7). This suggests to me that GAB sleep is 'deeper'. But I was confused by the comparison to the outcome of the behavioral experiments. There, they show that dFB-sleeping flies undergo more 'recovery sleep' than GAB-sleeping flies (Fig 5f), which I think suggests dFB sleep is 'lighter'. This would be consistent with the electrophysiology results, but the authors should please clarify if this is indeed their interpretation. Part of my confusion comes from the fact that the results are described as sleep 'after the red light is off' (line 249), but it sounds like they are really talking about sleep rebound.

One critical thing that is missing from Fig 5f is the control flies (and perhaps also sleep-deprived flies). Otherwise we cannot judge whether dFB flies are undergo more recovery sleep than normal flies, which would be hard to explain in their model. It would also be valuable to do the behavioral experiments with a combination of GAB- and dFB-induced sleep, to see if the double-induction blocks the effects on recovery sleep, which is what the LFP results predict.

Other important points:

Line 75: Authors state that flies slept readily on the ball. They should clearly state their definition of sleep at this point in the Results section. This is integral to everything that comes next in the paper. (Although there is a consensus definition in the field, I couldn't actually find a clear definition of what they call a sleep bout versus a wake bout anywhere in the paper. It should be stated clearly in the Results so it's easy for the reader to find).

Minor:

Many figures zoom tightly in on the 7-10Hz frequency band (Fig 2b, 3e,i, 5b etc.) I found it more revealing to see a broader display that stretches perhaps from 1-40Hz, like that in Fig 1. Otherwise we can't judge if other frequencies are changing or if it's specific to a relatively narrow band.

Line 158: 'Flies with only their brains thus heated...' It doesn't appear that the authors ever measure the temperature of other parts of the fly, so they should avoid stating this as though the temperature change is confined to the head.

Fig 1:

Authors should say in the text of the Results (not just the Methods) what time window/threshold they take to determine a response. The question hits you when you see the top example in Fig 1b, where there clearly is movement after the stimulus, but not immediately. So how stimulus-locked does the movement have to be?

The time axis on the bottom panel of Fig 1b is in the opposite direction to the time axis on the top panel. It would be better to put them both in the same direction. The direction in the top panel makes the most sense to me.

Fig 1e: What does n=10 refer to, the number of flies recorded or the number of sleep/wake bouts?

Fig 5e

This panel very unclear to me. Are these box plots? What are the bars within the boxes for 'GAB Day' and 'Control Day' and why are they not present in the other boxes? Why do all the error bars (SEM or SD or outliers or ??) extend to the same point (0)?

Both reviewers were positive about the study, which we have improved by acting on all of their helpful comments. Answers to each their comments are in blue, below.

Reviewer #1 (Remarks to the Author):

From behavioral and electrophysiological perspectives, sleep is not a homogenous state, and consists of multiple sub-states, with potentially different functions. This is well-established in mammals but less so in invertebrates, such as fruit flies. The van Swinderen lab previously described different sleep states associated with different levels of arousal threshold in van Alphen et al (2013). Here, they expand upon those findings by performing an array of electrophysiological experiments in vivo. The take-home message of this work is that a 7-10 Hz oscillatory signal seems to identify a transitional stage of sleep, which is seen in flies with natural sleep as well as dFB-activated sleep. In contrast, sleep induced by gaboxadol is not significantly associated with this 7-10 Hz signal. While this manuscript does not necessarily provide major conceptual advances, it provides rigorous data from technically challenging approaches. Because this is largely a descriptive paper, it is important to more fully characterize this 7-10 Hz oscillation associated sleep.

We hope that our reworked text and additional behavioral analyses help convince the reviewer that there is a conceptual advance as well.

Major comments:

1. In wild-type sleeping flies, the authors should assess arousal thresholds associated with sleep with greater vs reduced 7-10 Hz oscillations. Instead of shaking, the authors could use air puffs to test arousal threshold in this experiment. One might expect that the sleep associated with more 7-10 Hz oscillations to be lighter.

To address this criticism, we examined recordings from a separate group of wild-type flies which we stimulated every 15min throughout the night. Our strategy here was to capture moments (by chance) when the arousing stimulus (a mechanical vibration delivered to the fly holder, as in Figure 1) occurred at the same time as a 7-10Hz oscillation, to see if responsiveness was different then. This way, we were indeed able to probe many spontaneous 7-10Hz epochs, and compare these to sleep epochs without any 7-10Hz oscillation. We chose to use a vibration stimulus rather than an air puff as proposed because we could easily adapt the same method (as in Figure 1) to address this question.

We were somewhat surprised to find no difference in 7-10Hz in flies that were responsive versus unresponsive during sleep (see Figure R1, below; we also saw no significant effects in all other frequency domains, not shown here). But in retrospect, this is actually consistent with our dFB vs Gaboxadol arousal threshold data (now in a new Figure 6): dFB-induced sleep is NOT 'lighter' than Gaboxadol-induced sleep. We appreciate that this does not follow easily from behavioural experiments presented by van Alphen *et al* (2013), which

showed that flies are on average more responsive soon (0-5min) after falling asleep, compared to later 'deep sleep'. One explanation could be that not all of the first five minutes of early sleep are comprised of this 7-10Hz oscillation; there could still be non-oscillatory lighter sleep happening, on average. We did not know if/when 7-10Hz activity is happening in our first behavioral studies, which tracked average responses in fly populations. However, when we now specifically query whether responsiveness during sleep is associated with increased levels of 7-10Hz activity, we see no significant effect (Figure R1). While the usable sample size is small (these were difficult experiments), we are confident that this conclusion is the most likely. We have now clarified this in the paper to avoid confusion: 7-10Hz activity describes a different form of sleep, but this does not necessarily mean it is lighter. We have removed such language about lighter and deeper sleep stages from the revised text.

Figure R1. Averaged 7-10Hz LFP power (ratio to wake LFP) within 1 min prior to arousal stimulus, split into trials when the flies responded and when the flies did not respond to the stimulus (n=6, ns, Wilcoxon matched-pairs signed rank test, two-tailed).

Related to this issue, we have since submitted another manuscript where we investigate further dFB activation and related effects on behavior. We conclude in that paper that dFB activation acutely blocks behavioral responsiveness, so a logical link to the current study is that blocking of responsiveness promotes an effective entry into sleep, before other neurochemical pathways take over to maintain arousal thresholds at a high level. In this regard, it makes sense that arousal thresholds might be as high during this dFB-mediated signal which blocks behavioral responsiveness. We can provide the results from this related publication upon request. Also, we prefer not to include Figure R1 in the paper, as these remain inconclusive results for the brain recording preparation; but we could fit it in if requested.

2. I am interested in the macro-architecture for sleep related to this 7-10 Hz signal. The authors show that within a single sleep bout that the 7-10 Hz signal is greater entering and exiting sleep. What about at a 6 or 12 hr level? For example, in mammals, there is greater SWS in the 1st part of the night and more REM in the latter part of the night. Is there generally more sleep with 7-10 Hz signal in the late night compared to the early night?

This was an excellent suggestion. We reexamined our data to address this question, by dividing the night time sleep bouts into two epochs: early sleep and late sleep, or first third of sleep versus last third of sleep. While we noted much more variability in 7-10Hz activity in early sleep, we did not detect any significant differences with 7-10Hz during late sleep. These data are summarized below in Figure R2. This suggests that the 7-10Hz oscillation is not homeostatically regulated. We have now included this interesting

speculation and the result in the paper, as a new Supplementary Fig. 1c.

Figure R2. Normalized LFP power for the 7-10Hz domain of the first third of night sleep compared to the last third of night sleep. Wilcoxon matched-pairs signed rank test, two-tailed, N=10). No significant differences were found.

3. Is 7-10 Hz associated sleep regulated by circadian or homeostatic forces? One may get a sense of this by providing data for the above point. However, to directly address this, the authors should also repeat this in clock mutants and also following 12 hours of sleep deprivation.

As suggested, we addressed this question by partitioning our sleep data between early and later recordings during the night. As outlined above, we found no significant differences in 7-10Hz power. This was surprising, but we now report the result in the text.

We agree that looking at certain clock mutants could be very interesting, but feel that such work would call for an additional level of control experiments (e.g., subjective day and night, genetic controls) that would require a tremendous amount of more work beyond the scope of our current study. To still address this concern however, we bit the bullet and performed the additional analysis on a dataset of flies that has been mechanically perturbed. These were flies that had been stimulated every 15min throughout the night, thus sleep deprived: if there were any homeostatic effects, they should be evident here, especially later in the night. Consistent with our statistical analysis of the first unstimulated dataset (Figure R2), we found no statistical differences in 7-10Hz power in this mechanically disturbed group (Figure R3). We debated whether to include these data as a new panel in the paper, but felt that it does not add much more than what we already added with our re-analysis of the first dataset. We therefore prefer to leave the figure flow as is. We understand however that a better answer to this question will require a much larger effort (e.g., 10 sleep-deprived flies versus 10 controls), which could be the beginning of an entire paper focused on exactly what aspects of brain LFP activity are under homeostatic control. We will do this in the future.

Figure R3. Normalized LFP power for the 7-10Hz domain of the first third of night sleep compared to the last third of night sleep, for a group of flies that were mechanically perturbed every 15min. Wilcoxon matched-pairs signed rank test, two-tailed, N=7). No significant differences were found.

4. Overall, I did not find Figure 5 helpful, as I do not think that combining Gaboxadol with dFB activation yields particularly useful information since both drug dose and thermogenetic activation can be varied. I was also confused by the arousal threshold data in Figure 5e. It seems that the arousal threshold is greater when dFB neurons are activated compared to when Gaboxadol is administered, while I was under the impression that Gaboxadol-induced sleep is deeper. Also, there is very little signal range in Figure 5e (particularly at night)—the authors should try stronger stimuli to bring out potential differences in arousal threshold.

We have taken this advice on board and created an entirely new figure (Figure 6) to add new data and also better explain why these behavioral readouts are important for the paper. By the same token, we have expanded the electrophysiological analysis in the old Figure 5 to better show why combining both sleep manipulations is important for understanding that these probably engage distinct sleep stages in flies.

In the new Figure 5, we now incorporate some of the supplementary electrophysiology data into the main figure. Since this was previously relegated to the supplement, readers may have missed that light-induced dFB activation after Gaboxadol exposure does not activate the brain at all, for any frequency. This result allowed us to conclude that these pathways are unlikely to be parallel in the brain, and that dFB-induced sleep depends to some extent on the GABAergic sleep processes induced by Gaboxadol. An alternative result might have been that dFB-induced oscillations still occur after Gaboxadol-induced sleep, which might have suggested a parallel mechanism unrelated to sleep. That is clearly not the case, and we hope the new Figure 5 helps clarify that.

We felt that it was important to elaborate on the behavioral consequences of either sleep manipulation, during but also after prolonged sleep induction with either method. This is now all in the new Figure 6. We agree that our previous figure might have caused some confusion: the consequences of either sleep manipulation were investigated using sleep intensity as a readout, so sleep was used to study sleep effects, which may be problematic. We now also examine responsiveness in awake flies, to provide a more valid functional readout for our different sleep manipulations. We also add sleep deprivation data to better compare whether either of our sleep manipulations are

depriving key sleep functions. Not surprisingly, sleep deprivation decreases behavioral responsiveness afterwards in awake flies (this is a new finding), along with increasing sleep intensity (this is consistent with our previous work). We found that after having forced dFB sleep, flies then still need to sleep almost as deeply (during the subsequent day) as sleep-deprived flies. What was most surprising was that, unlike sleep-deprived flies, their behavioral responsiveness levels (% responding) while awake was normal. This suggests that prolonged dFB sleep accomplishes some sleep functions, namely for maintaining normal levels of behavioral responsiveness to mechanical stimuli, but not other sleep functions – as evident by a homeostatic deep sleep rebound. It is interesting to speculate that dFB sleep has failed to fulfill other sleep functions such as stress regulation and metabolite clearance, but it is of course too premature to say this in our paper. In contrast, we see no such effects following Gaboxadol-induced sleep, nor for combined Gaboxadol and dFB sleep. These results are consistent with our electrophysiology showing that Gaboxadol suppressed dFB oscillations. We feel that our behavioral data strengthen our story, to better convince that dFB sleep is doing something different. We hope the reviewers agree, and appreciate how their criticism forced us to come up with a new, more relevant behavioral readout: responsiveness in awake flies. It makes more sense now.

Minor comments:

1. In figure 1, when electrodes were inserted into the optic lobes, did the authors confirm electrode placement by post-hoc labeling?

This was done in a subset of the flies; mostly insertion site was estimated stereotactically, as in previous publications using the same preparation (e.g., van Swinderen, 2007).

2. Can the authors provide more details on how their Logitech webcam was modified to detect infrared signal?

We have added text to the Methods explaining this, in lines 599-607.

Reviewer #2 (Remarks to the Author):

The authors' main claim is that they have identified different phases of sleep in *Drosophila*, which seem to 'have different effects on behavior' (line 24). They establish this by recording field potentials in flies whose movements (and thereby sleep state) are simultaneously measured by their rotations of a spherical treadmill, and showing that rhythmic activity in a particular frequency band (7-10Hz) is more prominent at the beginning and end of a sleep bout (Fig1g). This is quite a nice technique, and the use of an electrode targeted specifically to the Fan-shaped Body (a brain area implicated in sleep) in Fig3 is a good addition to the relatively non-directed use of a multi-electrode array.

Overall the paper is a reasonable contribution to the literature. The first part of their claim, the different sleep phases, is reasonably convincing. However, the second part, what role different phases play behaviorally doesn't have a strong answer: the authors just show that recovery sleep is different after two

different ways of inducing sleep (Fig 5f). What that accomplishes is not addressed. But the first part is still a valuable contribution, and I think the second part is largely a matter of changing the language to tone down that aspect of their claims. If the authors could change the language to avoid the implication that they have shown these phases of sleep are 'functionally distinct' (e.g. in the Abstract, changing the section heading 'Behavioral Effects...' to Behavioral correlates), and generally making it clear that their evidence stops at behavioral correlates, the statements would be more justified.

Please see our comments to Reviewer#1 above regarding the behavioral analyses now presented in a new Figure 6. We feel more confident about these conclusions now, and hope the Reviewer agrees. We have nevertheless also tried to adjust our language in the revised manuscript, while taking care to still convey the relevance of our findings.

I have two major critiques:

1) The experiment showing an increase in 7-10hz power with dFB-induced sleep needs larger n (Fig 2b and Fig 3g). In Fig2b, for the Trp/104y flies, I cannot see why the only significant effect observed is the middle red bar, and not also the middle blue bar (post-heat), based on the magnitudes and the error bars. If anything the magnitude of the increase in power is larger post-heat than heat-on. I think the authors need to dig into the statistics here and see what's the basis for the significance. They should also increase the n since there's absolutely no reason why channels 1-5 and 11-15 are so different from one another in the Trp/104y panel (compare top and bottom blue bars; also true but smaller effect comparing top and bottom red bars). Clearly these differences must come from noise. Given that those noise-based differences are quite large, I think they need much more than n=7 to reliably detect the effect they are interested in.

The main goal of these exploratory experiments was to determine if two previously-reported sleep-promoting lines (104y and C5 x UAS/TrpA1) revealed any changes in LFP activity upon activation, and if so, whether this activity was widespread or localized in the brain. As such, we feel that these experiments served their key purpose: LFP activity is significantly increased when these circuits are activated, and these effects occur primarily in the central brain. These results guided our subsequent FB-targeted recordings.

However, we acknowledge that the data and statistics were not presented in the best way. There are several aspects to Fig. 2 that may have caused confusion. First, in regards to Fig. 2 'Heat Off' being non-significant, this comparison was actually significant at an alpha of <0.05 (0.048) however, it did not reach the adjusted alpha level (<0.025) after Bonferroni correction for which reason we did not mark it as significant. Second, we realize there was also a discrepancy between the data displayed in Fig. 2 and the data the statistics were based upon. The data displayed in Fig. 2 was based on the medians. The statistics however were based on the means (median over channel, mean over time) – due to baseline correction if we had taken the median over time as with the bar plots, the baseline condition would have

been nothing but zeros which would not have been appropriate for statistical comparison as we were interested in comparing to baseline with a repeated measures approach. Furthermore, owing to the positive skew in the dataset and the presence of some noise, we felt medians allowed the data to be represented more clearly. As we employed a non-parametric form of ANOVA which converts the response variables to percentiles to deal with skew, we felt the medians might be more appropriate than means. This may not have been clear in the text so we have added an explanation about the aligned-rank ANOVA that we used, in the Methods.

In response to reviewer concerns listed elsewhere, we have adjusted Figure 2 to include a wider range of frequencies to better represent the findings of our analysis, including frequencies between 2-40Hz instead of only 7-10Hz. We note that now the 'heat off' as well as the 'heat on' period is statistically significant compared to baseline for this frequency band. That there might be some inertia here would not be inconsistent with these being sleep-promoting circuits. We hope that the new presentation of Figure 2 (and its associated supplemental figure) is now better aligned with the exploratory nature of these experiments. Although we concede that the sample sizes are low, these are nevertheless significant effects (the statistics are now better explained in the Methods). Altogether, data from over 20 flies thus allowed us to conclude that: 1. Sleep promoting circuits *increase* LFP activity in the brain, and 2. Increased LFP activity is mostly in the central brain. We believe that is enough to convince the reader why we then proceeded to the next logical step: sharp recordings from the central brain, during various sleep manipulations.

The FB-targeted recordings in Fig 3 are a better technical approach than the trans-brain recordings in Fig 2. But again in Fig3g, the Trp/104y flies 'Heat On' show a very small effect, but what is noteworthy is that the Post Heat bars for these flies show very low variability, lower than the +/104y bar, and lower than the OL recordings in Fig3j. The fact these control/no effect conditions fail to show a consistent result again suggests that they haven't yet got a large enough sample size.

We admit the sample sizes are on the low side for some of our data. This reflects the necessary removal of entire datasets contaminated by heartbeat artifacts (this was surmised independently by two people, M.Y. and M.J.G., explained in the Methods). However, having internal controls helps. The statistical comparisons are against baseline (pre-heat), and our statistical methods are appropriate. The high level of variability for the 'heat on' condition reflects the fact that not all flies responded to dFB activation in the same way. We do not know why the 'post heat' condition has comparatively such low variance – perhaps once sleep has been induced, LFP variance in this specific frequency domain is low afterwards. Not all flies respond to dFB activation the same way. The reason there is high variance in the 104/+ (control) central brain recordings may be a response to the heat stimulus in awake animals. This is one reason we moved on to optogenetics later in the paper, to see if a similar LFP increase could be evoked using a different activation procedure, namely light. The use of optogenetics was also crucial

for better accomplishing our final 'combined' sleep manipulations, as temperature shifts are not a good strategy when doing any pharmacology.

The critical piece of information showing that there are different states during normal sleep is the plot in Fig 1g, indicating power in the 7-10hz band fluctuates. It is important the authors show a plot similar to Fig1g for GAB-induced sleep to be able to make a direct comparison. Clearly there are still some 7-10hz bouts in their data with GAB-sleep (they point them out in Fig4g). They never directly test whether the those bouts are still more prominent at the beginning & end of sleep epochs in the GAB-induced sleep.

None of our Gaboxadol experiments increased LFP activity, on average. Only 0.2mM Gaboxadol had any significant effect on LFP activity, and that was to *decrease* LFP activity, across all frequency domains. This effect is now better dissected (across frequency domains and sleep duration) in a new Supplementary Figure 7. The reviewer will appreciate here that, while some sleep time differences may still be evident for some frequency domains, the main effect of Gaboxadol is to decrease LFP activity. For these data, the exact moment of awakening was less clear than for spontaneous sleep bouts, and this may be reflected in our finding that late sleep was still significantly lower than wake. We believe adding these analyses was a useful suggestion, as it now better shows how different Gaboxadol-induced sleep is than dFB-induced sleep. This is now also supported with new behavioral data (Figure 6).

Finally, I suggest moving the experiments with the other dFB driver in Sup Fig 4 to the main Fig 2. 104y clearly labels many non-dFB neurons, so having a partly overlapping driver in c5 is very important to increase confidence the effect is really derived from the FB expression.

As mentioned previously, we have adjusted Figure 2 to include frequencies between 2-40Hz to better show our findings with the multichannel preparation. C5 was the noisier of the GAL4 lines tested with the multichannel, hence our subsequent focus on 104y. But we understand that adding the C5 data provides confidence in the results. We felt that the best way to do this was to provide the average spectrograms side by side in Supplementary Figure 4, where it is clear that both manipulations dramatically increase LFP activity. We prefer to keep this comparison in the supplement, so that the main figure can be focused on the strain we selected for subsequent experiments. We have provided a Supplemental Text further explaining these results.

2) With the LFP recordings, when they combine GAB- and dFB-induction of sleep, the oscillations they observe from dFB go away (Sup Fig 7). This suggests to me that GAB sleep is 'deeper'. But I was confused by the comparison to the outcome of the behavioral experiments. There, they show that dFB-sleeping flies undergo more 'recovery sleep' than GAB-sleeping flies (Fig 5f), which I think suggests dFB sleep is 'lighter'. This would be consistent with the electrophysiology results, but the authors should please clarify if this is indeed their interpretation. Part of my confusion comes from the fact that the results are described as sleep 'after the red light is off' (line 249), but it sounds like they are really talking about sleep rebound.

We agree that this may have been confusing in the previous version of the manuscript. We have now separated our final set of results into two sections: electrophysiology and behaviour. With electrophysiology (new Figure 5), we hope it is clear that dFB-related LFP activity is not possible under Gaboxadol-induced sleep. This supports our claim that these are not parallel pathways, and that dFB-induced sleep is in a way superseded by an alternate form of sleep. What was indeed surprising is that the former is not 'lighter' than the latter, as now shown more clearly in a new Figure 6. However, the consequences of either sleep manipulation are not the same (as outlined above). This suggests that either sleep induction method on its own may achieve different functions, because dFB sleep still requires as much of a deep-sleep rebound as sleep deprivation (while still preserving wakeful responsiveness levels). We realize however that the functional relevance here is still highly speculative, and have adjusted our language and conclusions accordingly. The best we can do now is compare to the effects of sleep deprivation (these are new data), where it is clear that prolonged dFB sleep looks like sleep deprivation in some ways (e.g., subsequent sleep intensity).

One critical thing that is missing from Fig 5f is the control flies (and perhaps also sleep-deprived flies). Otherwise we cannot judge whether dFB flies undergo more recovery sleep than normal flies, which would be hard to explain in their model. It would also be valuable to do the behavioral experiments with a combination of GAB- and dFB-induced sleep, to see if the double-induction blocks the effects on recovery sleep, which is what the LFP results predict.

These were two excellent suggestions that we followed, which provided us with clear results and a stronger ending for the paper. Greatly appreciated.

Other important points:

Line 75: Authors state that flies slept readily on the ball. They should clearly state their definition of sleep at this point in the Results section. This is integral to everything that comes next in the paper. (Although there is a consensus definition in the field, I couldn't actually find a clear definition of what they call a sleep bout versus a wake bout anywhere in the paper. It should be stated clearly in the Results so it's easy for the reader to find).

We have now included our defining criterion early in the paper as suggested.

Minor:

Many figures zoom tightly in on the 7-10Hz frequency band (Fig 2b, 3e,i, 5b etc.) I found it more revealing to see a broader display that stretches perhaps from 1-40Hz, like that in Fig 1. Otherwise we can't judge if other frequencies are changing or if it's specific to a relatively narrow band.

We have zoomed out for figures where we agree this would help better convey the effects we have found (e.g., 2-15Hz in Fig. 3e,i). Elsewhere, we

have kept the narrow focus if that was truly the only area showing effects.

Line 158: 'Flies with only their brains thus heated...' It doesn't appear that the authors ever measure the temperature of other parts of the fly, so they should avoid stating this as though the temperature change is confined to the head.

This is a fair point. We have adjusted our text accordingly.

Fig 1:

Authors should say in the text of the Results (not just the Methods) what time window/threshold they take to determine a response. The question hits you when you see the top example in Fig 1b, where there clearly is movement after the stimulus, but not immediately. So how stimulus-locked does the movement have to be?

We have now clarified this in the Results (line 82) as well as the Methods.

The time axis on the bottom panel of Fig 1b is in the opposite direction to the time axis on the top panel. It would be better to put them both in the same direction. The direction in the top panel makes the most sense to me.

Good point. We have fixed this.

Fig 1e: What does n=10 refer to, the number of flies recorded or the number of sleep/wake bouts?

N refers to the number of flies. We have now clarified this in all the Legends.

Fig 5e

This panel very unclear to me. Are these box plots? What are the bars within the boxes for 'GAB Day' and 'Control Day' and why are they not present in the other boxes? Why do all the error bars (SEM or SD or outliers or ??) extend to the same point (0)?

We have clarified the panels in the new Figure 6, and in the legend. These are indeed boxplots, with the central bar representing medians. All flies were equally unresponsive (NR = Non Responsive) at night, so differences in arousal threshold are only detectable during the day. This also highlights the fact that induced sleep intensity is as strong as spontaneous sleep at night (although it probably isn't doing exactly the same thing in the case of dFB).

Reviewers' comments:

Reviewer #1 (Remarks to the Author):

The authors do an adequate job of addressing the concerns of the reviewers, although in some respects their revisions have raised more questions than provided answers. At this point, I have a couple of suggestions for improving the manuscript. I still think the paper is a helpful contribution to the literature, but some issues need to be clarified so that readers are not misled.

1. One relative weakness of the paper is that it is not clear what the function of this 7-10 Hz associated sleep is. Although it appears to be transitional, the authors report that re-analysis of their previous data suggests that it is not associated with the reduced arousal threshold seen in "lighter sleep." Consistent with this observation is their re-analysis suggesting that this type of sleep is not homeostatically regulated. I think it is important to show the data that 7-10 Hz oscillation-associated sleep is not associated with changes in arousal threshold. It is also important to discuss this in the context of their previous van Alphen et al (2013) findings, because I think many readers might assume that this transitional sleep is lighter sleep and may be misled.

2. I am confused by the comment re: their recently submitted manuscript suggesting that dFB activation may block behavioral responsiveness. If this is the case and if 7-10 Hz oscillatory sleep is an important related to dFB-induced sleep, then shouldn't there be a difference in arousal threshold with 7-10 Hz oscillation-associated sleep? This raises the issue of what aspects of dFB activation are related to 7-10 Hz oscillations. I suspect that dFB activation engages multiple processes (one of which may be 7-10 Hz oscillatory activity), but that this 7-10 Hz activity may not be relevant for the dFB-induced behavioral data that the authors provide. Moreover, there is no clear increase in 7-10 Hz activity when using a cleaner dFB driver, potentially consistent with this notion. The authors should be careful to dissociate their findings related to 7-10 Hz activity and their dFB-activation behavioral data. Although there is a temptation to link them to provide smoother logic and seamless intellectual appeal, I think it is potentially misleading.

3. Regarding the relationship between dFB-induced sleep and Gaboxodal-induced sleep, their data are consistent with a model whereby dFB-induced sleep acts upstream of Gaboxodal. This is not surprising, since one would expect global activation of GABA_A receptors to be downstream. However, in terms of comparing the whether dFB-induced sleep or Gaboxodal-induced sleep is "deeper" (Fig. 6d), I would be careful comparing these manipulations, because they are qualitatively different (neural activation vs drug intervention) and, as I mentioned before, both manipulations are tunable.

4. On page 16 in the discussion, they speculate that 7-10 Hz oscillatory activity may be similar to sleep spindles that may block processing of external stimuli, allowing for a higher arousal threshold. I suggest removing the phrase about "arousal threshold." Again, I am concerned about trying to link the dFB activation data (with the greater arousal threshold) and their 7-10 Hz oscillation associated sleep data.

Reviewer #2 (Remarks to the Author):

The paper is much improved. I particularly like the new Fig 6 which shows some behavioral consequences of the two different types of sleep (dFB- versus GAB-induced). My concerns have been adequately addressed.

Two small things to mention:

Fig 1c far right 'Wake' zoom panel. The LFP trace appears to be clipped at top and bottom. Please adjust axis range.

Figure 1d is not referred to in the main text. Please either devote a sentence of the main text to the panel or move it to the Supp so it doesn't interfere with the narrative (doesn't seem essential to me).

Reviewer #1 (Remarks to the Author):

The authors do an adequate job of addressing the concerns of the reviewers, although in some respects their revisions have raised more questions than provided answers. At this point, I have a couple of suggestions for improving the manuscript. I still think the paper is a helpful contribution to the literature, but some issues need to be clarified so that readers are not misled.

1. One relative weakness of the paper is that it is not clear what the function of this 7-10 Hz associated sleep is. Although it appears to be transitional, the authors report that re-analysis of their previous data suggests that it is not associated with the reduced arousal threshold seen in “lighter sleep.” Consistent with this observation is their re-analysis suggesting that this type of sleep is not homeostatically regulated. I think it is important to show the data that 7-10 Hz oscillation-associated sleep is not associated with changes in arousal threshold. It is also important to discuss this in the context of their previous van Alphen et al (2013) findings, because I think many readers might assume that this transitional sleep is lighter sleep and may be misled.

We agree that this is an important point to make. To address this, we have added an additional panel, Fig. S1d, showing that responsiveness to mechanical stimuli is not significantly different when 7-10Hz LFP activity is high versus low, during tethered sleep. This required some reanalysis of our arousal-probing dataset (outlined in the Methods).

We believe the function of 7-10Hz sleep is to ‘block’ behavioral responsiveness while flies transition to sleep states under stronger neuromodulatory control. It then would make sense why arousal thresholds should be as high for this form of ‘active’ sleep, as it is for sleep states characterized by lower LFP activity. This of course does not answer yet what kind of brain activity is associated with lower-arousal threshold sleep, as described in van Alphen et al 2013. We suspect that this might involve sleep epochs when LFP activity has not yet reached its lowest point, but which also does not involve increased 7-10Hz activity.

We now make sure that these caveats are clearer in the last revised version of the manuscript, and have added relevant text to the results, discussion, and methods (in blue) accordingly.

2. I am confused by the comment re: their recently submitted manuscript suggesting that dFB activation may block behavioral responsiveness. If this is the case and if 7-10 Hz oscillatory sleep is an important related to dFB-induced sleep, then shouldn't there be a difference in arousal threshold with 7-10 Hz oscillation-associated sleep? This raises the issue of what aspects of dFB activation are related to 7-10 Hz oscillations. I suspect that dFB activation engages multiple processes (one of which may be 7-10 Hz oscillatory activity), but that this 7-10 Hz activity may not be relevant for the dFB-induced behavioral data that the authors provide. Moreover, there is no clear increase in 7-10 Hz activity when using a cleaner dFB driver, potentially consistent with this notion. The authors should be careful to dissociate their findings related to 7-10 Hz activity and their dFB-activation behavioral data. Although there is a temptation to link them to provide smoother logic and seamless intellectual appeal, I think it is potentially misleading.

Please see the reply to the related issue, above. It is true that optogenetic activation of the cleaner dFB circuit (23E10) did not recapitulate the clean 7-10Hz oscillations we saw during spontaneous sleep (or, indeed during 104y/TrpA1 activation, Supplemental Movie 1). We do

not know why this is the case, and agree that some more cautionary language is appropriate here. We have now added this (in blue), in the results and discussion.

3. Regarding the relationship between dFB-induced sleep and Gaboxodal-induced sleep, their data are consistent with a model whereby dFB-induced sleep acts upstream of Gaboxodal. This is not surprising, since one would expect global activation of GABA_A receptors to be downstream. However, in terms of comparing the whether dFB-induced sleep or Gaboxodal-induced sleep is “deeper” (Fig. 6d), I would be careful comparing these manipulations, because they are qualitatively different (neural activation vs drug intervention) and, as I mentioned before, both manipulations are tunable.

We agree, and have added a final line at the end of the results expressing this important point.

4. On page 16 in the discussion, they speculate that 7-10 Hz oscillatory activity may be similar to sleep spindles that may block processing of external stimuli, allowing for a higher arousal threshold. I suggest removing the phrase about “arousal threshold.” Again, I am concerned about trying to link the dFB activation data (with the greater arousal threshold) and their 7-10 Hz oscillation associated sleep data.

Done, as requested.

Reviewer #2 (Remarks to the Author):

The paper is much improved. I particularly like the new Fig 6 which shows some behavioral consequences of the two different types of sleep (dFB- versus GAB-induced). My concerns have been adequately addressed.

Two small things to mention:

Fig 1c far right ‘Wake’ zoom panel. The LFP trace appears to be clipped at top and bottom. Please adjust axis range.

This has been corrected, as requested, by adjusting the scale.

Figure 1d is not referred to in the main text. Please either devote a sentence of the main text to the panel or move it to the Supp so it doesn’t interfere with the narrative (doesn’t seem essential to me).

We did refer to Fig. 1d in the text, it was in (current) lines 88-90:

We used wavelet analysis²¹ to examine how LFP frequencies changed through time, across 24 hours of wake and sleep. As found previously^{11, 19, 20}, sleep in flies is associated with overall decreased LFP activity (Fig. 1c,d; Supplementary Fig. 1b).

We realise this was pretty subtle, and easy to miss. However, we are basically just replicating previously published results here, which is why we don’t dwell on this. Still, we feel it is important to make clear that these increased 7-10Hz oscillations are occurring in a backdrop of overall decreased LFP activity during fly sleep, as is also evident in the sample spectrograms (Fig. 1c).

REVIEWERS' COMMENTS:

Reviewer #1 (Remarks to the Author):

The authors have addressed my remaining concerns, and I have no further questions. As I mentioned before, I think this work will be a valuable addition to the literature.